# Studying Synaptic Connectivity and Strength with Optogenetics and Patch-Clamp Electrophysiology

**DOI:** 10.3390/ijms231911612

**Published:** 2022-10-01

**Authors:** Louisa E. Linders, Laura. F. Supiot, Wenjie Du, Roberto D’Angelo, Roger A. H. Adan, Danai Riga, Frank J. Meye

**Affiliations:** Department of Translational Neuroscience, Brain Center, UMC Utrecht, Utrecht University, 3584 CG Utrecht, The Netherlands

**Keywords:** patch-clamp electrophysiology, optogenetics, brain slices, CRACM, synapses, plasticity, connectivity, dual color optogenetics

## Abstract

Over the last two decades the combination of brain slice patch clamp electrophysiology with optogenetic stimulation has proven to be a powerful approach to analyze the architecture of neural circuits and (experience-dependent) synaptic plasticity in such networks. Using this combination of methods, originally termed channelrhodopsin-assisted circuit mapping (CRACM), a multitude of measures of synaptic functioning can be taken. The current review discusses their rationale, current applications in the field, and their associated caveats. Specifically, the review addresses: (1) How to assess the presence of synaptic connections, both in terms of ionotropic versus metabotropic receptor signaling, and in terms of mono- versus polysynaptic connectivity. (2) How to acquire and interpret measures for synaptic strength and function, like AMPAR/NMDAR, AMPAR rectification, paired-pulse ratio (PPR), coefficient of variance and input-specific quantal sizes. We also address how synaptic modulation by G protein-coupled receptors can be studied with pharmacological approaches and advanced technology. (3) Finally, we elaborate on advances on the use of dual color optogenetics in concurrent investigation of multiple synaptic pathways. Overall, with this review we seek to provide practical insights into the methods used to study neural circuits and synapses, by combining optogenetics and patch-clamp electrophysiology.

## 1. General Introduction

Ex vivo brain slice preparations allow for the electrophysiological study of the functional intricacies of the synapse. Over the years these approaches have given crucial insights in the functional architecture of neural circuits. Interrogating synaptic function in brain slices has classically relied on the evaluation of spontaneously occurring neurotransmission (i.e., spontaneous and miniature postsynaptic currents) and of neurotransmission evoked through electrical stimulation of axons (i.e., evoked postsynaptic currents). While these approaches remain in use and powerful today, an inherent problem with them is that for most central synapses they do not permit exact determination of the presynaptic origins of the recorded synaptic activity. 

Two critical findings helped overcome this limitation. First, the discovery that transfecting mammalian neurons with algae-derived light-sensitive ion channel channelrhodopsin-2 (ChR2), confers light-driven control over excitability of such neurons at the soma level [1,2]. Second, the discovery that in vivo transfection of neurons with ChR2 also leads to its functional presence at axons and nerve terminals [3]. This latter feature permits the study of the function of specific synapses in a brain slice, without requiring the cell bodies of the stimulated presynaptic axons to remain present in it. The overall approach involving: (a) the expression of ChR2 (or a variant) in presynaptic neurons, (b) preparing brain slices, (c) patch-clamping postsynaptic cells, and (d) evaluating postsynaptic currents evoked by light-driven stimulation of the presynaptic side, was originally termed “ChR2-assisted circuit mapping” (CRACM) [3]. 

CRACM allows for the more precise study of the architecture of neural circuits, but also for the evaluation of differences in pre- and postsynaptic functioning. Indeed, patch clamp electrophysiology has a long history of specific metrics that can be taken to study separate features of synaptic function, and many of these measures are compatible with optogenetic approaches. This review gives an overview of how CRACM-like approaches are used in practice to establish crucial parameters of neural circuit function. We discuss how such approaches are used to determine the presence of ionotropic, but also of metabotropic types of synaptic connectivity. We discuss the rationale for and meaning of distinct synaptic functionality metrics. Finally, we examine the status quo regarding the use of dual color optogenetics to study multiple synaptic pathways ex vivo. Overall, we provide, to the relative beginner in such approaches, practical insight in the usage of optogenetics-assisted patch clamp approaches and we discuss relevant caveats. Here, we will mainly discuss how these approaches can be used to study (in rodents) neural circuits linked to the processing of valence information (i.e., reward and punishment). However, the principles we discuss are generalizable across circuit types, and the approaches can also be applied in multiple vertebrate and invertebrate species. 

## 2. Optogenetically Assessing Synaptic Connectivity

When studying neural circuits, a fundamental question pertains to whether or not there is synaptic connectivity between sets of neurons. Electrophysiological techniques remain the gold standard to determine this, allowing one to: establish functional synaptic connections, assess the presynaptically released neurotransmitters, and unravel the types of postsynaptic receptors involved. Whereas patch-clamping allows for the selection of the postsynaptic cells under investigation, optogenetic approaches allow selection of the presynaptic inputs assessed. After presynaptic targeting of the optogenetic actuator (often by stereotactically injecting viral vectors such as Adeno-Associated viruses (AAV) into a brain region [4,5]), several weeks of incubation are typically required for its full expression at nerve terminals (e.g., 3–6 weeks) [6]. Before brain slices are prepared, and prior to the patching process, there are several methodological considerations to be made that influence the protocols used. First, the type of synaptic connectivity that is assessed (e.g., glutamatergic or GABAergic). Second, the class of postsynaptic receptors mediating synaptic communication (e.g., ionotropic receptors or metabotropic G protein-coupled ones). Third, the source of synaptic input, since aside from monosynaptic connections it is also possible to measure indirect polysynaptic connections in a brain slice. Depending on the experiment at hand, this latter feature may either be an asset to exploit or a potential confound to be aware of. Below we discuss ways in which CRACM-like approaches can be used to determine these distinct features of synaptic connectivity in an input-specific manner.

### 2.1. Characterization of Synaptic Connectivity via Ionotropic Receptors

A feature of ionotropic receptor transmission is that it involves synaptic currents with relatively fast kinetics (Figure 1A). Typically, these synaptic responses can be observed when delivering a single time-locked light pulse to the brain slice, either via LED or laser. For ChR2-based experiments, or for other blue light sensitive variants, it is common practice to deliver brief blue light pulses (~470 nm wavelength pulses of 1–20 ms duration, often in the 1–75 mW/mm^2^ irradiance range). Tens of recorded sweeps with such time-locked light stimulation, typically with minimally 10 s between the sweeps, will then be averaged and synaptic response features such as the average amplitude are analyzed [1,6,7]. To detect the presence of various types of synaptic receptor type currents, there are important general patch-clamp considerations to make with regard to the internal pipette medium used (which will quickly dialyze the internal environment of the patched cell [8]; Box 1) and the potential at which the cell is clamped in case of whole cell voltage-clamp approaches (Box 2). We discuss these considerations for experiments to assess different types of excitatory or inhibitory ionotropic synaptic connections.

Box 1Considerations for internal pipette medium.For pipette solutions for whole cell patch recordings of synaptic activity, there are (at least) two major questions to address regarding the salt that forms the basis of the medium:
Is potassium or cesium the major cation?Is chloride or an alternative (e.g., gluconate or methanesulphonate) the major anion?1. High intracellular potassium concentrations represent a physiological situation. Nevertheless, in many experimental settings potassium may be replaced by cesium. Cesium ions block various potassium channels of the patched cell, reducing its leakiness, improving its impedance [9,10]. There are advantages of this compared to using potassium-based internals. First, cesium-based internals allow for relatively better clamping of cells near potentials that deviate strongly from their resting membrane potentials (also see Box 2). Second, cesium-based internals allow for improved detection of (optogenetically evoked) synaptic inputs that impinge onto the cell further away from the pathed site (typically the soma). Nevertheless, even with cesium-based internals considerable attenuation of the amplitude of synaptic inputs onto distal dendrites occurs [9,10]. Potassium-based internal solutions have other advantages. While they still allow for detection of (proximal) optogenetically evoked synaptic responses, they are compatible with acquiring additional measures from the cell such as action potential properties. They also allow for detection of synaptic transmission that is based on potassium fluxes (e.g., certain metabotropic receptor currents; Section 2.2; Figure 1B).2. In mature neurons low levels of chloride ion concentrations in the cell represent the physiological situation. Nevertheless, experimenters often choose to use an intracellular medium with a high chloride concentration. There are advantages to this compared to using low chloride internals. First, with low concentrations of chloride in the cell GABA_A_R transmission is typically (weakly) inhibitory when the cell is near resting membrane potential (chloride flowing into the cell). Instead, when intracellular chloride concentrations are high such GABA_A_R transmission becomes strongly excitatory (chloride flowing out of the cell) (Figure 1A). A high chloride internal solution thus provides better signal-to-noise ratio when the goal is to detect GABA_A_R-mediated synaptic currents, though at the cost of its directionality typically being inverted from the physiological situation (Figure 1A). Second, using high chloride internal solutions diminishes ionic composition differences between the artificial cerebrospinal fluid (ACSF) and the pipette medium, facilitating clamping conditions. Ion species can differ in mobility and when ACSF and pipette medium differ in their major anion type (e.g., chloride in ACSF and gluconate in a low chloride internal) liquid junction potentials occur which affect voltage clamping values [11]. With chloride salts in the internal (i.e., same main anion in the internal and in the ACSF), liquid junction potentials will typically be close to 0 mV, not affecting voltage clamp values. Overall, internal solutions each have their own strengths and weaknesses and their use needs to be guided by the requirements of the experiment.

#### 2.1.1. Considerations for AMPAR-Mediated Synaptic Connectivity

Optogenetics combined with patch clamp is often used to study the presence of glutamatergic synaptic contacts from a given input source onto the patch clamped neurons. To this end typically the occurrence of optogenetically-evoked AMPA receptor (AMPAR)-mediated currents in patched neurons is assessed. Experimenters often use internal solutions high in cesium ions (Box 1) to record optogenetically evoked AMPAR-mediated currents [12,13,14,15]. In whole-cell voltage clamp configurations the direction and magnitude of a given synaptic current is determined by the (clamped) potential across the membrane (i.e., electrical driving force) and by the intracellular versus extracellular concentrations of the ion species to which the receptors underlying the synaptic currents are permeable (concentration gradient) (Box 2). Regardless of the internal solution opted for, voltage clamping values typically used to determine the presence of AMPAR-currents are in the range of −60–75 mV. At such potentials the driving forces (Box 2) governing ionic current flow through open AMPARs are such that they result in strong “inward currents” (i.e., defined in patch clamp terminology as either the influx into the cell of positively charged ions, or as the efflux of negatively charged ones) primarily driven by influx of sodium ions into the cell. 

#### 2.1.2. Considerations for NMDAR-Mediated Synaptic Connectivity

When performing connectivity experiments, one needs to also consider NMDAR-mediated synaptic currents. This is especially the case since these may occur in the absence of AMPAR-mediated ones at so-called silent synapses, which can be created but also unsilenced by salient experiences. For instance, environmental stimuli such as addictive drugs [16,17] or stress [18] can ultimately drive AMPAR insertion in the postsynaptic membranes of previously silent synapses. In the absence of AMPAR responses it is therefore possible to miss (the potential for) ionotropic glutamatergic synaptic connectivity if NMDAR responses are not evaluated. To allow for detection of NMDA-mediated synaptic events, recorded cells are typically voltage-clamped at depolarized potentials (e.g., in the vicinity of +40 mV), to alleviate the voltage-dependent magnesium block of the NMDAR [19], at which these synaptic events manifest as outward currents. In this case, cesium-based internal solutions are required to ensure sufficiently adequate voltage clamping of the cell at sites where NMDAR-driven glutamatergic inputs are received (Box 1). An alternative strategy that can rely on potassium-based internal solutions, uses magnesium-free extracellular artificial cerebrospinal fluid (ASCF), to alleviate magnesium-block of the receptor in that manner. With that approach, optogenetically driven inward NMDAR currents can be observed with voltage clamping nearer to resting membrane potentials [20].

#### 2.1.3. Considerations for GABA_A_R-Mediated Synaptic Connectivity

When studying ionotropic GABA_A_R-mediated currents there are several considerations to make in terms of pipette solution and holding potentials. With internal solutions that seek to maintain a relatively physiological level of both the potassium and the chloride concentration in the (mature) neuron (e.g., potassium gluconate; Box 1), GABA_A_R-responses can be detected as inhibitory “outward” responses (i.e., mainly Cl^−^ flowing into the cell) when the neuron is clamped at slightly depolarized values (e.g., −50 mV) (Figure 1A; Box 2) [21]. Clamping the neuron at much more hyperpolarized potentials than that can be problematic as it brings the cell closer to the GABA_A_R reversal potential, which is typically in the −70 to −80 mV range, and is determined by mainly Cl^−^ and to a lesser extent HCO3^−^ conductance (Box 2). At this potential no net synaptic current would flow upon GABA binding the receptor, such that clamping near these potentials would make it easy to miss GABA_A_R-dependent connectivity. Clamping the neuron at more depolarized potentials to distance the cell further away from the GABA_A_R reversal potential is problematic for other reasons. First, it may result in occasional action potential generation in the cell that can obscure the detection and confound the interpretation of synaptic currents. Depending on the cell type this may already be an issue in the range of −50–60 mV. In this case, one can consider adding the sodium channel blocker QX-314 to the internal solution, which blocks action potential generation in the patched neuron. More problematic however is that it is not practically possible to clamp the neuron near such depolarized potentials using a potassium-based internal (Box 1). For these reasons, alternative approaches regarding the internal medium are often taken. With a cesium-based internal with low chloride concentrations (e.g., cesium methanesulfonate) one gains the ability to clamp the patched cell at much more depolarized potentials (i.e., farther away from the reversal potential, Box 2) that allow for easier detection of outward GABA_A_R currents. Alternatively, one can use internal solutions with a high Cl^−^ concentration (e.g., potassium chloride or cesium chloride) such that near resting membrane potentials this results in strong “inward” GABA_A_R currents (i.e., chloride efflux), permitting detection of GABA_A_R events with a high signal-to-noise ratio (Figure 1A; Box 1). 

Box 2Considerations for clamping potentials: Driving forces and reversal potentials for receptors mediating synaptic currents.To understand the direction and driving forces of synaptic currents it is important to understand the reversal potential for the synaptic receptor current: the membrane potential at which there would be zero net current flow across the receptor if it were to be put in an open state (i.e., neurotransmitter-bound). If the neuron has a membrane potential more depolarized than the reversal potential (E) for a synaptic receptor type, then the synaptic current flowing through that receptor (e.g., I-GABA_A_R or I-AMPAR) will drive the neuronal membrane potential towards the reversal potential for that receptor (e.g., E_GABAaR_ or E_AMPAR_). Synaptic receptor currents tend to be mediated by multiple ion species. Most AMPARs (and NMDARs) in an open state are non-selective cation channels through which prominently Na^+^ and K^+^ ions (but also cesium) can flow [22]. Instead GABA_A_Rs are permeable most prominently to Cl^−^ ions, but also to a lesser extent to HCO3^−^ ions [23]. Reversal potentials for synaptic receptor currents can be approximated by the Goldman-Hodgkin-Katz (GHK) equation that takes multiple ionic components and different receptor permeabilities for them into consideration [22,23]. Although the concentration of extracellular (ACSF) and intracellular (pipette medium) are made by the experimenter with known concentrations, there are still often uncertain factors at play (e.g., the exact intracellular concentration of HCO3^−^ is often not known) that may complicate exact GHK-based calculations. The reversal potential for a receptor can however be empirically determined by finding a voltage clamping value where the synaptic current, when (optogenetically) evoked, leads to zero current.Since between labs there are often subtle differences in patch clamp recording conditions (e.g., in the exact ionic composition and pH of the internal solution and the ACSF used) it is not possible to give one generalized reversal potential for a given synaptic receptor current. However, typical reversal potentials for I-AMPARs and I-NMDARs are around 0 mV [12,22]. For I-GABA_A_Rs, when using low chloride internal solutions, reversal potentials tend to be in the range of −70 to −80 mV. If a cell is voltage clamped at for instance −50 mV, I-GABA_A_R will be mainly dominated by chloride anions flowing into the cell: signifying a hyperpolarizing “outward” current (Figure 1A top) [23]. Instead, if internal solutions with high chloride concentrations are used, E_GABAAR_ will be shifted to much more depolarizing values, such that upon GABA_A_R transmission chloride anions will strongly flow out of the cell, signifying a depolarizing “inward” current (Figure 1A bottom). Note that the terminology of inward and outward currents from a voltage-clamp perspective relates not to direction of general ion flux, but rather “inward current” refers to either cations flowing into the cell or anions flowing out of the cell, and “outward currents” refer to the opposite scenarios.

### 2.2. Characterization of Synaptic Connectivity via Metabotropic Receptors

Aside from detecting fast ionotropic signaling, CRACM approaches can be used to assess current responses generated by postsynaptic metabotropic receptor activation. Compared to ionotropic current responses, metabotropic receptor-mediated currents tend to depend on downstream effector channels (e.g., G protein-coupled inwardly rectifying potassium channels, GIRKs), have slower kinetics and often require trains of light stimulation pulses delivered at relatively high frequencies (e.g., >20 Hz) to be revealed [24,25,26] (Figure 1B). 

Trains of optical pulses delivered at middle to high frequencies (e.g., 5–50 Hz) have been successfully used to examine this type of synaptic connectivity. In one study, outward GABA_B_R-mediated currents were evoked by ChR2-driven optogenetic stimulation of entopeduncular synaptic inputs to the lateral habenula (LHb) [24]. Habenular neurons were voltage clamped at −50 mV with a potassium gluconate internal solution and ChR2-expressing entopeduncular inputs were optogenetically stimulated with pulse trains. Trains of 10 pulses each, given at 5, 10 or 20 Hz, resulted in both fast ionotropic GABA_A_R outward currents as well as slow outward currents that were blocked by a GABA_B_R antagonist. The amplitude of the GABA_B_R-mediated current increased with higher frequencies of optical stimulation [24]. Another study similarly evaluated GABA_B_R transmission at nucleus accumbens (NAc) synapses onto ventral tegmental area (VTA) dopamine neurons [27]. VTA cells were voltage clamped at −55 mV, using a potassium methanelsulfonate internal, and ChR2-expressing NAc inputs were optogenetically stimulated with trains of 20 pulses, delivered at 20 Hz. The resultant outward currents were also blocked by a GABA_B_R antagonist [27]. 

Neuropeptidergic signaling via G protein-coupled receptors (GPCRs) has also been investigated using optogenetic approaches. In one study orexin transmission in the hypothalamus was evaluated between ChR2-expressing orexin neurons and patch-clamped histamine neurons [25]. Histamine neurons were clamped at −70 mV with a potassium gluconate internal, and orexin axons were light stimulated with 30s 20 Hz pulse trains. This evoked slow inward currents in the histamine neurons that were blocked by an orexin 2 receptor antagonist [25]. Another study in the hypothalamus performed ChR2-mediated stimulation of neurokinin-B releasing arcuate nucleus neurons [26]. In this study current clamp recordings were made from the same ChR2-expressing neurons with a potassium gluconate internal. Light stimulation with (≥5 s) trains of pulses of 10, 20, 30 or 50 Hz (but not with 1 or 5 Hz) produced slow excitatory postsynaptic potentials (EPSPs) after the stimulation train ended. This slow EPSP was blocked by an antagonist for Tacr3, a GPCR for neurokinin-B [26]. As a final example, a recent study showed that optogenetic stimulation of VTA dopamine neurons somato-dendritically released the neuropeptide cholecystokinin (CCK) [28]. VTA dopamine neurons expressing ChR2 were patch clamped (potassium chloride internal, current clamp) and were stimulated with 1 s trains of 20 Hz for a period of 6 min. This resulted in local CCK release, as established by an immunoassay, and such protocols potentiated GABA_A_R-mediated inputs onto dopamine cells via CCK2 receptors [28]. 

Overall, various forms of synaptic connectivity mediated via direct metabotropic receptor-mediated currents can be studied using optogenetic approaches in conjunction with patch-clamp electrophysiology. As metabotropic responses often rely on downstream engagement of potassium channels, the use of potassium-based rather than cesium-based intracellular solution, which block potassium channels, are often necessary when evaluating these types of current (Figure 1B; Box 2). Metabotropic-mediated postsynaptic currents typically require moderate to higher frequency trains of stimulation. 

### 2.3. Characterizing Monosynaptic vs. Polysynaptic Connectivity 

When optogenetic stimulation of presynaptic inputs results in a detected synaptic response in a patched neuron, this can reflect a direct synaptic pairing (i.e., monosynaptic connectivity; (Figure 1C). Alternatively, intermediate neurons in-between the optogenetically stimulated neurons and the recorded ones, could mediate the ultimately measured current (i.e., polysynaptic connectivity; Figure 1D). A typical example of polysynaptic connectivity that can be detected in a brain slice involves a glutamatergic input impinging on a downstream cell by disynaptic feedforward inhibition (i.e., stimulating nearby local GABAergic interneurons that in turn connect with the patched cell; Figure 1D). 

To differentiate between mono- and polysynaptic connectivity, often the different temporal features of these distinct types of synaptic responses are used. Such features include the latency of onset of the optically evoked postsynaptic currents after stimulation (Figure 1D). In addition, the variance in response latency (i.e., the response jitter) is typically different between monosynaptic (small variance) and polysynaptic (larger variance) responses [29,30]. However, disynaptic inputs can also occur quite fast [30,31]. For instance, for excitatory medial prefrontal cortex (mPFC) inputs synapsing onto basolateral amygdala (BLA) pyramidal cells, the onset of monosynaptic glutamatergic currents was ~4 ms after the start of the light pulse. Although slower, disynaptic feedforward inhibition still resulted in synaptic currents with an onset of less than 7 ms [31]. Since the architecture of neural circuits differs with regard to where on the neuron the synapses for both mono- and disynaptic inputs occur, it is likely not feasible to arrive at precise guidelines across circuits based on the time of response onsets. Thus, it is challenging to reliably separate between monosynaptic and polysynaptic responses purely on strict temporal features. 

A less ambiguous approach to validate that a synaptic response is monosynaptic or not involves the use of the pharmacological cocktail of the voltage-gated sodium channel blocker tetrodotoxin (TTX) and the potassium channel blocker 4-Aminopyridine (4-AP). TTX blocks action potentials and abolishes both mono- and polysynaptic (opto-evoked) neurotransmission. In the case of a monosynaptic response, the amplitude of the opto-evoked synaptic transmission will be (at least partially) rescued by the addition of 4-AP, which prolongs the depolarization of local presynaptic terminals, boosting vesicle fusion and release [32,33]. TTX and 4-AP-assisted pharmacological isolation of monosynaptic connectivity (Figure 1C–E) has been applied to a plethora of brain regions and pathways in recent years [26,27,31,32]. For example, the aforementioned study on mPFC to BLA mono- and disynaptic communication, showed that optogenetic stimulation of mPFC terminals resulted in excitatory input to BLA pyramidal cells [31]. This was identified as a monosynaptic input, as its TTX-induced abolishment was reversed by the addition of 4-AP to the extracellular recording solution. Instead, mPFC stimulation also resulted in (slower onset) GABA_A_R-mediated outward currents in the BLA pyramidal cells. These were identified as polysynaptic inputs, as their abolishment by TTX was not restored by 4-AP [31]. It is notable that also metabotropic receptor monosynaptic connectivity can be revealed with the combination of TTX and 4-AP, for instance for GABA_B_R connectivity [27] and for neurokinin-B Tac3 receptor signaling [26].

### 2.4. Summary and Caveats

Combining optogenetics and patch-clamp recordings of neurons in brain slices allows for the detection of synaptic connectivity and for its characterization in terms of involved receptor types. When optogenetic actuators are virally expressed it is important to wait a sufficient amount of time after viral delivery of the opsin to permit optimal presence at axonal end points, prior to making brain slices [6]. Certainly when the point of the experiment is to assess if there is connectivity between neuronal subsets. Waiting times of at least 3 weeks, but also often even more than 6 weeks are commonplace [15,34].

It is important to keep in mind that since these approaches are performed in a brain slice, it is always possible that certain parts of a dendritic arbor of a patched cell on which synaptic inputs would occur are lost during the preparation. Moreover, even when using cesium-based internals when patching neurons at the soma level, there will still be (remote) parts of the dendrite impervious to voltage clamping and where impinging inputs are subject to strong filtering. Consequently, synaptic responses at very distal inputs may be missed [9,10]. If connectivity is found, some caution is also warranted in overinterpreting the magnitude of opto-evoked synaptic current amplitudes. While they can provide insight in the strength of a connection, amplitudes are strongly influenced by how many axons are being opto-stimulated. This may differ across brain slices, and across animals with variability in the extent to which virally targeted brain regions express the opsin. Without normalization procedures (also see Chapter 3) the characterization of synaptic connectivity therefore often takes a qualitative description (e.g., there is a monosynaptic inhibitory GABA_A_R-mediated connection between regions A and B).

Overall, optogenetics combined with patch clamp electrophysiology allows for establishing the type of connectivity at play if synaptic connectivity is indeed found. Nevertheless, the absence of connectivity in these types of experiments, particularly in regard to polysynaptic connections, should be cautiously interpreted as potential evidence for actual absence of any connectivity. 

## 3. Using Optogenetics and Patch-Clamp to Assess the Function and Strength of Specific Synapses

Aside from using optogenetic approaches with patch-clamp electrophysiology to determine synaptic connectivity, an important objective is often to determine the (altered) strength of synaptic connections and the plasticity of synapses. Determinants of synaptic strength include both presynaptic and postsynaptic processes [35,36,37,38,39]. Presynaptic factors include the probability of release, the size of the readily releasable pool (RRP) of vesicles, and the amount of neurotransmitter in individual vesicles. Postsynaptic factors include the presence and functionality of different types of ligand-gated and metabotropic receptors (e.g., their number and subtypes) and the exact location of such receptors on the postsynaptic side (e.g., proximity to the presynaptic release sites) [35,36,37,38,39]. To assess changes in synaptic strength, a variety of electrophysiological patch clamp metrics can be taken. Here, we discuss these and address how optogenetics can be used to obtain them in an input-specific manner. 

### 3.1. AMPAR–NMDAR Ratios and I/V Relations of AMPARs

#### 3.1.1. AMPAR–NMDAR Ratios 

A typical question when assaying the effects of meaningful experiences (e.g., stress or reward) on glutamatergic synaptic strength, is whether the event has caused long-term potentiation (LTP) or depression (LTD) at excitatory synapses [39,40,41]. If feasible, this would be accomplished by a direct comparison of AMPAR-mediated responses (i.e., the ‘workhorse’ amongst ionotropic excitatory transmission) at stimulated synapses across brain slices for different experimental conditions. However, the magnitude of such responses strongly depends on how many axons are being stimulated and therefore varies greatly across recordings from different slices. In the case of electrical stimulation this depends on the exact location of the electrode on the slice, while in case of (wide field) optogenetic stimulation this often depends on differences between the extent of viral targeting of the brain region of interest with the opsin. While this may be expected to average out if sufficient amounts of recordings are made across slices and animals, it is still a likely source of considerable variation [42]. 

To provide partial normalization for the number of glutamatergic axons being stimulated in a slice, (opto)evoked AMPAR responses onto a cell are therefore often ‘normalized’ to the concomitantly observed NMDAR response on the same cell, obtained with the same axonal stimulation parameters. This is done on the assumption that AMPARs and NMDARs often co-occur at the same glutamatergic synapses. Whereas NMDARs are often particularly critical in the induction phase of synaptic plasticity, alterations in number and/or function of AMPARs at the synapse often underlie the expression phase of the plasticity [39,40,41]. The normalization of AMPAR to NMDAR responses offers a metric that is informative regarding the functional strength of the glutamatergic synapse, which is less dependent on methodological variability. There are multiple ways to determine AMPAR–NMDAR ratios (Figure 2A; Box 3). Most approaches involve cesium-based internals permitting better clamping properties at depolarized potentials (Box 1). Furthermore, most approaches require pharmacological blocking of other relevant postsynaptic receptors that may be engaged by single pulse axonal stimulation (e.g., GABA_A_Rs or nicotinic acetylcholine receptors [43]). 

Box 3Calculating AMPAR–NMDAR ratios.One common approach for measuring the AMPAR–NMDAR ratio is by taking the peak AMPAR response amplitude while voltage clamping the cell near resting or at slightly more hyperpolarized potentials (e.g., in the −60 to −70 mV range). This allows for a strong cation driving force through the AMPARs into the cell, generating inward currents (Box 2). Slower NMDAR currents typically do not occur at these potentials due to their magnesium block [19]. After having established the AMPAR response, the cell can then be voltage clamped at more depolarized potentials (typically at +40 mV), alleviating the magnesium block such that slow outward NMDAR responses co-occur with faster outward AMPAR currents. The NMDAR response is then either isolated pharmacologically from this dual current, to unambiguously detect its maximal amplitude (Figure 2A_1_) or the NMDAR current is evaluated at a delayed time point where AMPAR responses have already decayed (often in the 50–100 ms range after stimulation) [34] (Figure 2A_2_). A practical advantage of the latter approach is that it requires fewer pharmacological blocking steps during the experiment such that multiple measures can be taken across cells within a slice. A disadvantage is that the NMDAR amplitude is taken at a time point where much of its current has already decayed, and the residual current amplitude may be harder to detect. Moreover, NMDAR subtypes with distinct decay kinetics may impact on the measure, and for relatively fast decaying NMDAR subtypes this approach may not be feasible [44].Another approach is to use magnesium-free ACSF such that both AMPAR and NMDAR currents can be observed as inward currents at near resting membrane potentials (e.g., −60 mV to −70 mV range). For this approach a non cesium containing internal solution could be considered, as clamping potentials strongly deviating from resting membrane potential are not required in this case. A final, and commonly used approach is to measure both AMPAR and NMDAR directly at +40 mV as outward currents (Figure 2A_3_). A marked advantage of this is that both current types are measured at exactly the same kind of experimental conditions (e.g., the cell’s intrinsic membrane properties, such as its membrane resistance are constant). However, with this approach it has to be considered that certain AMPAR subtypes (e.g., GluA2 subunit lacking ones) are inwardly rectifying (see Section 3.1.2). Consequently, for such AMPAR subtypes a measure of their outward current amplitude (e.g., at +40 mV) is not the same as a measure of their inward current amplitudes [45,46]. Overall, it may, therefore, be informative to calculate AMPAR/NMDAR in multiple complementary ways when assessing synaptic strength changes.

#### 3.1.2. AMPAR I/V Relationships and the Rectification Index

Another normalized measure to assess AMPAR functionality pertains to how much current (I) at various resting membrane voltages (V) it conducts (i.e., the I/V relation of the AMPAR). This is informative as it reveals features about the subunit composition that make up the AMPARs, which have consequences for their ion conductivity (see below) [45,47]. Indeed, AMPARs are highly modulated homo- or hetero tetramers comprised of four types of subunits (GluA1-4). Whether AMPARs express the GluA2 subunit is critical with regard to its conductance and ion type permeability [45]. Most AMPARs in the adult brain have GluA2 subunits [45,46]. The vast majority of these undergo editing of the pre-mRNA resulting in an arginine (R) residue instead of the genetically encoded glutamine (Q) in a membrane pore loop region. The addition of this positively charged R residue to the pore, via such “Q/R editing” prevents entry of divalent ions like calcium through the AMPAR. Thus GluA2-containing AMPARs (with Q/R editing) are impermeable to calcium ions. Instead, GluA2-lacking AMPARs (or those containing GluA2 subunits without Q/R editing) are permeable to calcium [45]. GluA2-lacking AMPARs have a high single channel conductance for inward currents, making them more efficient drivers of neuronal excitability than their GluA2-containing counterparts [45,48]. However, at potentials more depolarized than the I-AMPAR reversal potential (typically 0 mV), GluA2-lacking AMPARs have strongly attenuated outward currents. This is due to the voltage-dependent block in the receptor pore by endogenous polyamines (like spermine). This effectively renders the GluA2-lacking AMPARs inwardly rectifying, i.e., relatively more inward current flows at hyperpolarized potentials compared to the outward current that can flow at depolarized potentials (Figure 2B; Box 4), making the I/V relation of the AMPAR non-linear, yielding higher rectification index values (Box 4). A change in subtype composition of AMPARs, for instance due to an experience like stress or reward, can signify potentiation of excitatory synapses and can be measured as higher rectification index values. Notably, since GluA2-lacking AMPARs are calcium permeable, their presence at the synapse can change the conditions under which synaptic activity patterns result in calcium-mediated synaptic plasticity [49]. 

Box 4Calculating AMPAR I/V relationships and rectification index.The I/V relation of AMPAR, which reflects whether or not they are rectifying, can be measured by recording opto-evoked glutamatergic input when the postsynaptic cell is sequentially clamped at (for instance) −60 mV, 0 mV and 40 mV, in the presence of NMDAR and GABA_A_R antagonists. Again, recordings at such various holding potentials are normally performed in cesium-containing internal solutions. With typical cesium internals, the reversal potential for AMPAR responses is around 0 mV (Box 2). The AMPAR rectification index (a measure of linearity of I/V relationships) can be calculated by dividing the peak of the synaptic inward response (e.g., the absolute current at −60 mV) by the peak of the synaptic outward response (e.g., the response at +40 mV which may be less taxing on the patched neuron than +60 mV). Often the rectification index is then normalized such that a value of 1 would indicate absence of rectification. In the example of determining the rectification index based on current peaks observed when clamping the cell at −60 mV and +40 mV, the resultant ratio can be divided by 1.5 (60 mV/40 mV), correcting for its relative position opposed to the reversal potential around 0 mV. In that case, if the rectification index is above 1 there is AMPAR inward rectification, suggestive of the presence of GluA2-lacking AMPARs. Researchers often add the polyamine spermine to their internal pipette solution to ensure that polyamine block will take place even if the patched cell is dialyzed with the internal contents of the patch pipette liquid [47], though there are also examples showing that adding polyamines is not (always) necessary to still observe AMPAR rectification [50,51].

#### 3.1.3. Practical Examples of Optogenetic Studies Investigating AMPAR/NMDARs and I/V Relations

Various studies performing optogenetics-assisted patch-clamp recordings have established that salient experiences can induce synaptic plasticity in specific neural circuits, as observed by alteration of the AMPAR–NMDAR ratio. For example, one study observed that repeated shock stress increased the AMPAR–NMDAR ratio at orbitofrontal cortex (OFC) synaptic connections onto BLA pyramidal neurons in subsequent days [34]. In this case, the ratio was calculated using the inward peak response at −70 mV for AMPAR-currents, and the outward response at +40 mV 60 ms post-stimulation was taken as the NMDAR current [34]. The authors proceeded to show that there was also enhanced AMPAR rectification at these synapses after stress by comparing optogenetically evoked excitatory postsynaptic currents (oEPSCs) at −60 mV and those at +60 mV. This suggested a higher presence of GluA2-lacking AMPARs after stress. This scenario was corroborated by demonstrating that in stressed animals (but not in controls), the OFC-BLA oEPSCs were partially reduced by bath-application of NASPM, a blocker for GluA2-lacking AMPARs [34]. 

Other examples come from the effects of stressful experiences on glutamatergic synaptic connections onto lateral habenula neurons. One study described that acute foot-shock stress resulted in diminished glutamatergic transmission onto habenula neurons, with AMPAR–NMDAR ratios decreased at optogenetically probed hypothalamic, pallidal, midbrain and extended amygdala inputs onto the habenular neurons [14]. In this case AMPAR/NMDAR was calculated using outward responses at +40 mV for both AMPAR and NMDAR, followed by pharmacological parsing of the two responses with an NMDAR antagonist. In further support of this reflecting a postsynaptic effect, the authors observed diminished presence of GluA1 AMPAR subunits in the habenula after the acute stress [14]. Further experiments showed that in stressed animals there was a reduction in the AMPAR rectification index at glutamatergic synapses onto habenular neurons, as well as a reduction in the sensitivity of EPSCs to the GluA2-lacking AMPAR antagonist NASPM. Overall, this study used optogenetic approaches to show that acute stress reduced GluA2-lacking AMPARs at various key glutamatergic inputs onto habenular neurons [14]. In contrast another stress protocol, namely chronic mild stress, has been shown to enhance the amount of GluA2-lacking AMPARs specifically at pallidal synapses onto lateral habenula neurons, based on an increased AMPAR rectification index [15]. Accordingly, the amplitude of AMPAR-mediated pallidal-habenular synaptic responses was more attenuated by GluA2-lacking AMPAR blocker NASPM in the chronically stressed mice. In this case, stress-driven differences occurred in the absence of AMPAR/NMDAR changes, calculated at +40 mV for both components [15], showing how these measures are partially dissociable. The optogenetic strategy used here permitted establishing that the GluA2-lacking AMPAR insertion after chronic stress occurred specifically at pallidal rather than at hypothalamic or midbrain glutamatergic inputs onto the habenular cells [15]. 

An illustration of how alterations in rectification index may impact on certain forms of AMPAR/NMDAR calculations comes from a study assessing the effect of cocaine self-administration on specific glutamatergic synapses onto NAc neurons with a dopamine 1 receptor (NAc_D1R_ neurons) [12]. Between amygdalar, hippocampal and mPFC inputs onto NAc_D1R_ neurons, only those from the mPFC exhibited a higher rectification index after cocaine experience. This suggests that prior cocaine experience resulted in the insertion of GluA2-lacking AMPARs at mPFC-NAc_D1R_ synapses. The authors also calculated AMPAR/NMDARs at these distinct synapses, using peak currents for both AMPAR and NMDAR at +40 mV, with pharmacological isolation of the AMPAR response with an NMDAR antagonist. Interestingly, this revealed a decrease in the AMPAR–NMDAR ratio at mPFC-NAc_D1R_ synapses, which had a higher rectification index. This example is particularly illustrative, as it highlights that a decreased AMPAR/NMDAR, when both components are measured at +40 mV, could reflect an underlying AMPAR subunit switch that makes the AMPARs inwardly (but not outwardly) more conductive (Figure 2B; Box 4). In accordance with these findings signifying strengthened glutamatergic synapses, cocaine experience increased the maximal inward EPSCs that could be optogenetically evoked at mPFC-NAc_D1R_ synapses [12]. 

#### 3.1.4. Summary and Caveats

AMPAR–NMDAR ratios and AMPAR I/V relationships give insight in the state of glutamatergic synaptic strength and the measures can be used when performing optogenetic synaptic interrogation. Although changes in AMPAR/NMDAR are often suggestive of changes in the AMPAR component, it is to be kept in mind that experience-dependent plasticity of NMDARs can also occur [49]. With a changing ratio it is important to corroborate which of the components, if not both, have actually changed. Relevant NMDAR-related changes can be in terms of NMDAR number, but also in subunit composition. The NMDAR is also a multi-subunit heterotetramer [19] and depending on the GluN2 subtypes present, the decay times of NMDAR currents can range from relatively fast (e.g., tau 50 ms) to 20-fold slower [44]. Therefore, when AMPAR–NMDAR ratios are calculated using post-stimulation timing to separate AMPAR from NMDAR, possible NMDAR subunit changes could impact on the metric. Even when the main change for an AMPAR–NMDAR ratio is indeed an AMPAR-driven change, the manner in which the AMPAR–NMDAR ratio is calculated can influence the interpretation. In particular, due to possible AMPAR subunit composition changes, AMPAR current sizes at depolarized potentials can be very different from at hyperpolarized potentials [12]. It is therefore often advisable to also assess the I/V relationship of the AMPARs in addition to the AMPAR–NMDAR ratio. Finally, it is important to note that whereas AMPAR/NMDAR changes are often interpreted as reflecting postsynaptic changes, other factors like the extent of presynaptic spill-over affecting extrasynaptic NMDARs may also affect the measure [52]. 

### 3.2. Paired-Pulse Ratios (PPR)

#### 3.2.1. How PPR Relates to Presynaptic Processes Including the Probability of Release

A common approach to study the presynaptic strength of ionotropic synaptic transmission is to measure the paired-pulse ratio (PPR; Box 5). PPR is generally considered to be inversely related to presynaptic vesicular release, such that synapses with low presynaptic vesicle release probability (i.e., a relatively weaker synapse) tend to have high PPR values and vice versa. An experience-dependent decrease in PPR at a synapse can reflect synaptic potentiation via enhanced presynaptic vesicle release probability, whereas an increase in PPR can reflect synaptic weakening due to decreased release probability [53,54,55].

Box 5Calculating Paired-Pulse Ratios.In the context of optogenetic synaptic interrogation the PPR protocol involves the rapid delivery of two light pulses separated by an inter pulse interval commonly in the 50–200 ms range. The PPR is calculated as the average amplitude of the second optogenetically evoked synaptic current (oPSC2) divided by the average amplitude of the first synaptic current (oPSC1; Figure 2C).PPR protocols identify whether the studied synapses exhibit paired-pulse facilitation (PPF, i.e., PPR > 1) or paired-pulse depression (PPD; PPR < 1). Typically, the shorter the inter pulse interval (i.e., time between the two stimulations), the larger the extent of the PPD or PPF. It may require intervals as long as several seconds before these forms of short-term plasticity are no longer observed (i.e., PPR = 1) [21,37]. Notably, synaptic facilitation and depression can also be studied using trains of pulses (>2) rather than just pairs, though the underlying molecular mechanisms may not be identical between the two processes [53]. Inter pulse intervals longer than 50 ms and with two pulses rather than longer trains may give better alignment of PPR values between optogenetic approaches compared to electrical stimulation [56] (see Section 5.1).

PPR often relates to release probability due to the nature of the readily releasable pool (RRP) of synaptic vesicles. The RRP is composed of a relatively small fraction (e.g., 5%) of the vesicles at an active zone that can be easily driven to exocytosis upon stimulation [37]. For synapses with a relatively high release probability, upon initial stimulation such a considerable portion of the vesicles of the RRP will be effectively released, that upon the quickly following second stimulation this leads to the release of fewer vesicles, resulting in PPD (PPR < 1). It has been proposed that the expected tendency of synapses is to exhibit depression, and that PPF mechanisms require more specialized processes [53]. The process of PPF is calcium-dependent. Upon initial stimulation of the presynaptic side strong local elevations of calcium occur through voltage-gated calcium channels (VGCCs), engaging low-affinity calcium sensors like synaptotagmin 1 or 2 for synchronous vesicle release [57]. The calcium in the nerve terminal diffuses within the nerve terminal, binds presynaptic calcium proteins, and temporarily forms a residual calcium signal. After a quick successive stimulation, the second calcium influx engages with the residual one to enhance release probability. Mediating this process is an interplay involving, among other factors, distinct presynaptic calcium sensors (i.e., various synaptotagmins) [53]. PPF mainly occurs at low release probability synapses, and often reflects a presynaptic phenomenon [53,58]. 

#### 3.2.2. Practical Examples of Optogenetic Studies Investigating PPR Differences 

There are multiple studies observing experience-dependent PPR differences with optogenetic approaches. For example, one study examined the effect of chronic restraint stress on mPFC-BLA synapses in mice. The authors showed that prior chronic stress exposure resulted in larger oEPSCs at glutamatergic mPFC-BLA synapses. Under control conditions these synapses exhibited PPF (PPR > 1; suggesting low release probability). Chronic stress reduced the PPR, suggesting a stress-driven increase in release probability at these synapses. In accordance with this scenario, there was also an increase in the frequency of glutamatergic neurotransmission events at these synapses [59]. Together these findings were in accordance with increased presynaptic glutamatergic output at these mPFC-BLA synapses after chronic restraint. 

Another study evaluated the effect of cocaine treatment on different NAc cellular subtypes with dopamine 1 (D1R) or dopamine 2 receptors (D2R) projecting to the ventral pallidum (VP) [60]. The authors observed that both NAc_D1R_ and NAc_D2R_ neuronal populations formed inhibitory synapses onto VP cells mediated by GABA_A_R currents. In saline control conditions both NAc_D1R_-VP and NAc_D2R_-VP synapses exhibited similar extents of PPD. Compared to saline control, cocaine pre-exposure had divergent effects on these initial PPRs, inducing a stronger PPD at the NAc_D1R_-VP synapses (i.e., reduced PPR, suggesting higher release probability), and a PPF at NAc_D2R_-VP synapses (i.e., increased PPR, suggesting reduced release probability). This suggested a cocaine-driven potentiation of NAc_D1R_-VP inhibitory synapses and a weakening of NAc_D2R_-VP inhibitory synapses. In accordance with this scenario the authors performed plasticity occlusion experiments, showing that in cocaine conditions NAc_D1R_-VP synapses could not be potentiated further with a high frequency stimulation protocol, whereas NAc_D2R_-VP synapses could not be further depressed by a low frequency protocol [60]. 

It can be relevant to compare PPRs across different synapses in microcircuits. This has been done for instance in the NAc, which receives excitatory AMPAR-driven glutamatergic synaptic inputs from various sources. One study observed that BLA inputs on NAc cells exhibited basal PPF, whereas ventral hippocampus (vHipp) and mPFC inputs exhibited basal PPD [61]. A subsequent study looked at the ability of monoamines like dopamine and serotonin to presynaptically modulate such specific glutamatergic synaptic inputs to the NAc. Bath application of serotonin decreased EPSC amplitudes at vHipp, BLA, and at paraventricular thalamic (PVT) inputs to the NAc. Instead, bath application of dopamine selectively decreased EPSCs at PVT-NAc synapses. In all cases where the monoamine application reduced EPSC amplitudes this coincided with increases in the PPR at these synapses, suggesting a decreased presynaptic release probability underlying the effects [13].

#### 3.2.3. Summary and Caveats

Changes in the PPR metric are often interpreted as altered presynaptic output, which may be due to underlying alterations in in the RRP size, differences in the extent of contributing synaptic active zones, and/or changes in presynaptic release machinery [37]. Notably, there are specific scenarios where PPR changes are not due to presynaptic but rather postsynaptic alterations. Among those are postsynaptic receptor desensitization, due to strong and continuous presynaptic stimulation, particularly following long trains of stimulation. Another potentially confounding factor is postsynaptic receptor saturation. For receptors with slow ligand-offset kinetics, some receptors may remain in a ligand-bound state after the first presynaptic stimulation such that when the second stimulation occurs, they are not yet available to participate in synaptic transmission. This can occur for instance for NMDAR-mediated transmission [58]. Both those processes would drive subsequent reductions in the synaptic response amplitudes to subsequent stimulations that would manifest as PPD, but originate from a postsynaptic mechanism. It is also important to note that direct light stimulation of nerve terminals expressing (calcium-conducting) opsins can result in different nerve terminal properties (and PPR values) than those achieved physiologically by action potentials upon axonal stimulation [56]. Overall, while a PPR change is a useful indicator of alterations of (pre)synaptic function, its attribution to presynaptic mechanisms such as release probability needs to be corroborated by other approaches. 

Notably, there is duality in how PPR and release probability link up to the concept of ‘synaptic strength’. A synapse with low PPR and presumably high release probability may be considered ‘strong’ when it comes to efficiently transmitting information with minimal axonal stimulation. Nevertheless, during repetitive axonal stimulation at high frequencies it may become more difficult for such a “depressing” high release probability synapse to keep transferring information with the same intensity due to the diminished synaptic response sizes occurring along the train. The PPR, in particular with trains of pulses, can therefore also be viewed as a metric for information gating at the synapse. For synapses with an initially low release probability a propensity for synaptic facilitation would act as a high-pass filter for information transfer. Only strong high frequency presynaptic stimulation, perhaps even in bursts, would generate sufficient vesicle release for large postsynaptic responses. Instead, depression at synapses can act as a low pass filter, limiting how much of a long bout of high frequency presynaptic activation continues to be translated into persistent postsynaptic responsivity [53].

### 3.3. Variance-Mean Analyses such as the 1/CV^2^ Metric 

#### 3.3.1. How 1/CV^2^ Relates to Presynaptic Processes including the Probability of Release

Many metrics of synaptic strength focus on average current responses, but the variance underlying those averages also holds profound information regarding underlying synaptic processes. ‘Quantal analysis’ (with a quantum referring to a discrete neurotransmitter-filled vesicle) refers to a group of statistics-based approaches that utilize such variance to approximate separate features of synaptic transmission (e.g., release probability of vesicles, their number of release sites, and their impact) [38]. An often used quantal analysis method assesses the coefficient of variance (CV) for the synaptic amplitudes, with CV = σ/μ, where μ is the mean synaptic amplitude and σ the standard deviation of those amplitudes (Figure 2D; Box 6). When recalculated as 1/CV^2^ this measure is proportional to mainly the presynaptic output of the synapse. If the variance of synaptic amplitudes is relatively low compared to the average synaptic response, this results in low CV values, meaning high 1/CV^2^ values, in accordance with high presynaptic vesicular output (Box 6). 

1/CV^2^-based analyses can be performed on optogenetically evoked synaptic currents from specific inputs. Often (but not exclusively) this is done within a recording where after a baseline of synaptic responses, a manipulation is performed to induce LTP or LTD. If the observed potentiation or depression occurs alongside increases or decreases in 1/CV^2^, respectively, the locus of plasticity is presumed presynaptic (i.e., alterations in release probability or number of release sites). If the synaptic amplitude changes occur while 1/CV^2^ remains constant, the plasticity expression locus is considered as likely postsynaptic (e.g., alteration in postsynaptic receptor levels, their positioning or conductivity) [62]. 

Box 6Calculating 1/CV^2^.Quantal analysis mathematically expresses the amplitude of synaptic transmission as the product of the amount of independent presynaptic release sites (n), the release probability of a single vesicle (p) and the impact in terms of current amplitude that a single vesicle has via postsynaptic receptors (q). Whereas ‘q’ makes up the ‘quantal size’, the product of ‘n’ and ‘p’ makes up the ‘quantal content’ [38,62,63]. Using binomial statistics, the mean peak average of a synaptic current (μ) is then assumed to be μ = n*p*q, whereas the variance (σ^2^) underlying this mean peak average is assumed to be σ^2^ = n*q^2^*p(1 − p). Since CV is operationally defined as σ/μ, that means that CV^2^ = (σ/μ)^2^ = (1 − p)/(n*p). Particularly relevant here is that CV (or 1/CV^2^) is thus considered to be explainable by presynaptically dependent factors p and n, and is not considered to be dependent on q, which is often presumed to be mainly dependent on postsynaptic factors. Often the CV metric is restated as 1/CV^2^ = (n*p)/(1 − p). First, this ensures that the metric is not affected by whether negative (e.g., inward AMPAR-mediated currents) or positive (e.g., outward GABA_A_R mediated) synaptic currents are analyzed. Second, this makes it such that 1/CV^2^ is theoretically proportional (rather than inversely so) to presynaptic release strength, as it will increase when n and/or p does (i.e., when the ‘quantal content’ increases). In specific scenarios 1/CV^2^ may even be mainly dependent on changes in p, if an evaluated synaptic transmission strength change occurs on a timescale assumed to be too fast for release site changes to have occurred (e.g., within several minutes) [62].

#### 3.3.2. Practical Examples of Optogenetic Studies Investigating 1/CV^2^ Differences

There are examples in which CV metrics and related variance-mean based analyses are performed using optogenetic stimulation. One study evaluated the effect of chronic pain (spinal nerve injury) on spike-time dependent plasticity at thalamic synapses onto anterior cingulate cortex (ACC) pyramidal neurons [64]. The authors observed that in control mice, pairing postsynaptic depolarization of the cortical neurons with subsequent optogenetic thalamic stimulation, resulted in LTD at these synapses. This LTD coincided with reductions in 1/CV^2^ (compared to the pre-LTD baseline). Instead, in the chronic pain group, spike time dependent LTD did not occur at these optogenetically probed thalamocortical synapses, and no changes in 1/CV^2^ were observed [64]. In this case, the 1/CV^2^ metric was taken within a singular recording (before and after LTD induction), such that the number of stimulated axonal release sites will have remained constant. Notably, the authors also found that basal (pre-LTD) levels of 1/CV^2^ were higher at thalamic-ACC synapses in the chronic pain group as compared to sham controls, as were the amplitudes of the opto-evoked responses (i.e., differences between recordings) [64]. Indeed, another study has also reported synaptic response 1/CV^2^ differences caused by a salient experience (i.e., between recordings). This study used CRACM approaches to investigate the state of mPFC projections to the BLA in wild type compared to Neurexin 1a knock out mice [65]. Glutamatergic mPFC-BLA synaptic AMPAR currents were smaller in the knockout mice. When light intensity was titrated to have similarly sized currents in WT and KO mice, the CV at PFC-BLA synapses was higher (i.e., a lower 1/CV^2^ value), which could be in accordance with reduced presynaptic output in the KO mice [65].

A more elaborate form of variance-mean analysis than 1/CV^2^ is multiple probability fluctuation analysis (MPFA). MPFA involves acquiring synaptic responses across various release probability conditions. This is followed by plotting synaptic variance against synaptic mean amplitudes, curve fitting this relationship, and extracting the estimated metrics from the typically parabolic relationship between variance and mean [38]. There are studies showing the potential for this approach with optogenetics. One study assessed whether variance-based analyses could be used between recordings to determine effects of cocaine withdrawal at mPFC-NAc synapses of rats [42]. The authors patch-clamped NAc neurons and applied optogenetic stimulation of five pulse trains (20 Hz). Different release probability was presumed to occur across the five pulses of the train, as there was synaptic depression along with accordant changes in CV^2^. The authors then used 30–100 of such 5 pulse trains to obtain parabolic variance-mean relationships. Subsequent MPFA analysis identified that at mPFC-NAc synapses there were specific increases in release probability in rats in cocaine withdrawal. In accordance with this, the PPR, which scales inversely with release probability (see Section 3.2), was lower in the cocaine withdrawal animals. Instead, based on MPFA analysis, the number of release sites stayed constant as did the quantal size. An advantage of MPFA over approaches like 1/CV^2^ or PPR, is that MPFA in theory approximates actual values for synaptic determinants (e.g., yielding an actual release probability value) rather than yielding parameters that scale with such processes. 

#### 3.3.3. Summary and Caveats

1/CV^2^ and related measures are metrics rooted in the binomial statistics used in quantal analysis to model synaptic functioning. There are certain assumptions underlying these approaches, which are not always fully met [38,62]. As the 1/CV^2^ measure is sensitive to the amount of release sites recruited, it is common to see applications of the metric at different phases during the same recording, when axonal stimulation settings remain relatively constant throughout. Nevertheless, the metric has also been used in optogenetic studies comparing basal 1/CV^2^ values between experimental groups [64,65]. There are some factors that may make optogenetic stimulation more adapted than electrical stimulation when it comes to comparing variance-mean metrics between recordings. First, the range in the number of axons and release sites that can be optogenetically stimulated in a brain slice tends to be smaller than the range that can be electrically stimulated. Second, whereas electrical stimulation tends to occur by placement of an electrode at a given spot in the brain slice (introducing variance in number of stimulated sites due to subtle placement differences), optogenetic stimulation often occurs with a wide-field configuration, limiting this type of experimenter bias [66]. In accordance with this, one study assessed thalamo-striatal synapses in brain slices, as probed with either electrical stimulation (by approximative electrode placement to target those pathways) or with optogenetics, and evaluated CV at these synapses in both cases. They observed that with optogenetics there was less dependency of CV values on the chosen stimulation intensity [67]. Even if optogenetic stimulation allows for interpretability of 1/CV^2^ differences across recordings in certain conditions, those measures should be done in conjunction with other supporting metrics to confirm mechanisms underlying any observed synaptic plasticity. Aside from 1/CV^2^ measures, analyses such as MPFA [38,42] and the variance-to-mean ratio (which depends on release probability and quantal size, but not the amount of release sites) [63] are interesting CRACM-applicable tools to unravel the locus of synaptic plasticity changes. Again, an important caveat of the optogenetic study of presynaptic metrics, is that in case of direct light stimulation of opsin-expressing nerve terminals, the terminal release properties can be different from those obtained with axonal stimulation [56].

### 3.4. Quantal Responses in an Input-Specific Manner Using Strontium in the Extracellular Medium

#### 3.4.1. How Strontium-Mediated Asynchronous Release Reflects Synaptic Quantal Sizes

The quantal size is the postsynaptic response elicited by a singular synaptic vesicle (Box 6). Synaptic quantal sizes are usually determined by taking the average amplitude of recorded miniature excitatory or inhibitory postsynaptic currents (mEPSCs and mIPSCs, respectively), in the presence of voltage-gated sodium channel blocker TTX. However, such spontaneously occurring synaptic responses are not input-specific and are therefore not easily reconcilable with CRACM approaches. As noted in Section 3.3, quantal analyses methods may provide tools to use input specific optogenetic approaches to gain insight in quantal sizes [38,42,63]. However, an alternative strategy to such statistics-based models involves the determination of quantal sizes by stimulation of axons in conditions that lead to asynchronous rather than synchronous vesicle release. Consequently, synaptic vesicles are not released at the same time but rather in succession. This can be accomplished with strontium replacement of the calcium in the ACSF (Box 7) and allows for the approximation of the impact exerted by an individual vesicle (in a “miniature”-like manner) for a specific optogenetically stimulated input (Figure 2E). The average amplitudes of these asynchronous quantal-like responses are typically taken as a proportional measure of synaptic strength, in a similar way as changes in mEPSC or mIPSC amplitudes are interpreted. 

To create conditions for asynchronous release in brain slices, typically the slice is submerged in extracellular medium in which calcium ions have been replaced with strontium ions (Box 7). Under regular calcium concentration conditions, depolarization of the presynaptic nerve terminal would lead to calcium influx via VGCCs to trigger synaptic vesicle release, both via a dominant phasic synchronous component and via a slower asynchronous component [57]. Under strontium-calcium replacement, depolarization of the nerve terminal leads to strontium ions entering the terminal through VGCCs [68]. Like calcium, strontium can drive both synchronous and asynchronous synaptic vesicle release, albeit considerably less efficiently than calcium does [69,70]. Strontium has lower affinity than calcium for the calcium sensors that mediate synaptic release [71]. Interestingly, the permeation of strontium ions through VGCCs in the presynaptic nerve terminal is larger than for calcium ions [70,71]. Moreover, strontium is more slowly extruded from the presynaptic nerve terminal upon stimulation [69]. Thus, upon presynaptic entry, the strontium ionic presence is elevated more persistently compared to calcium, but with lower binding to the type of presynaptic sensors that mediate synaptic release. Together these factors are considered to contribute to strontium driving protracted asynchronous quantal-like release [69,71].

Box 7Calculating strontium-mediated asynchronous quantal-like synaptic currents.Strontium-driven asynchronous release is achieved by replacing a typical calcium ion containing ACSF (e.g., 1.5–2.5 mM calcium ions) with a modified ACSF in which strontium ions (typically at a 2–8 mM concentration range) have replaced the calcium. Upon stimulation, a remnant of a synchronous response followed by hundreds of milliseconds of asynchronous release is  observed [21,61,69]. The amplitudes of those asynchronous events can be averaged to approximate an input-specific quantal size (Figure 2E). When performing such experiments, it should be validated that during a baseline period of similar length, there are substantially fewer events than in the post-stimulation period. Because the marked presence of those would mean considerable risk of ‘contamination’ of the input-specific quantal responses with non-input-specific synaptic events.

#### 3.4.2. Practical Examples of Optogenetic Studies Using Strontium to Assess Quantal Size Changes 

CRACM approaches can use strontium measures to help identify experience-dependent plasticity. For instance, in one study the effect of prior cocaine administration was evaluated on input-specific synapses onto patched neighboring NAc_D1R_ and NAc_D2R_ neurons [72]. The authors used strontium to demonstrate differences in asynchronous synaptic properties after cocaine pre-exposure, observing that cocaine exposure diminished the asynchronous event amplitude at vHipp-NAc_D1R_ synapses. In accordance with this being mediated by postsynaptic processes, the postsynaptic spine volume at vHipp-NAc_D1R_ synaptic sites was diminished in cocaine conditions [72]. 

Quantal amplitudes reflect not only the postsynaptic state, but also the content of the presynaptic vesicles, which can vary [73,74]. A study examined the effect of cocaine withdrawal in mice on glutamate/GABA coreleasing synapses from the pallidal entopeduncular nucleus (EPN) onto lateral habenular neurons. During withdrawal, the inhibitory GABA_A_R-dependent component of EPN-LHb synaptic transmission was diminished, thereby altering the excitation/inhibition balance at these synapses. The mechanism for this plasticity effect was identified as presynaptic, involving reduced presynaptic GABAergic vesicle filling. Accordingly, in conditions of strontium replacement of calcium, decreased amplitudes of asynchronous synaptic events (i.e., the impact of individual vesicles released) were observed during cocaine withdrawal [21].

Aside from evaluating the amplitudes of asynchronous events it can be informative to consider their frequency. A previously mentioned study observed that chronic restraint stress in mice resulted in strengthening of glutamatergic mPFC-BLA synapses [59]. In accordance with a presynaptic contribution to the stress-driven synaptic potentiation, PPR was decreased after stress. Moreover, in strontium ACSF conditions optogenetic stimulation of the synapse resulted in a larger frequency (but not amplitude) of asynchronous events in stressed mice [59]. 

#### 3.4.3. Summary and Caveats

Overall, the application of strontium allows approximation of quantal sizes in an input-specific manner. When interpreting the amplitudes of the asynchronous events, it is important to consider that they can reflect postsynaptic (e.g., receptor presence), but in certain instances also presynaptic factors, particularly since the neurotransmitter content of presynaptic vesicles is not fixed [21,73,74]. Therefore, in case of observed differences in asynchronous amplitudes, corroborating measures need to be taken to identify the synaptic locus of effect. While it is also possible to evaluate frequencies of strontium-driven asynchronous events, it is important to consider that these will also depend on the number of axons and release-sites stimulated, which is more difficult to keep comparable between recordings.

### 3.5. Optogenetic-Assisted Study of Opioid G Protein-Coupled Receptor Control over Specific Synapses 

Thus far, we have discussed various CRACM-applicable metrics that can be taken to assess the state of synaptic strength. Another major determinant in setting the parameters of synaptic function is the action of metabotropic GPCRs, which are not only present on postsynaptic elements, but are also powerful regulators of presynaptic functionality across valence circuits [75]. A prominent example of a subfamily of GPCRs in the context of valence processing are opioid GPCRs [76]. We will briefly discuss how optogenetics with brain slice patch clamping can be used to assess the synapse regulatory role of the mu-opioid receptor (MOR), delta opioid (DOR), and the kappa opioid receptor (KOR) on specific inputs in a brain slice context. These opioid GPCRs all belong to the G_i/o_ family and are located at both presynaptic axon terminals and postsynaptic sites [77]. The endogenous ligands for such opioid receptors are released upon strong stimulation procedures. In the amygdala, metenkephalin (an opioid with affinity for DORs and MORs) was found in dense core vesicles within axon terminals that terminated either on other axon terminals or on postsynaptic sites [77]. Typically trains of high frequency stimulation or prolonged neuronal depolarization are required to release these opioids [77,78]. Aside from axonal release, endogenous opioids may also be somatodendritically released [78]. Stimulated presynaptic opioid GPCRs typically suppress vesicle release, while stimulating postsynaptic opioid GPCRs can drive potassium efflux, often via GIRKs, to cause neuronal hyperpolarization [77,79,80,81].

#### 3.5.1. Practical Examples of Optogenetic Studies of Opioid GPCR Control over Specific Synapses

Optogenetics together with patch-clamp electrophysiology and slice pharmacology is a potent way to determine which specific synapses are modulated by (opioid) GPCRs and how. For example, one study used such an approach to determine that KORs inhibit specific glutamatergic synapses onto NAc_D1R_ but not NAc_D2R_ neurons in a pathway-specific manner [80]. The authors evaluated the sensitivity of BLA and vHipp glutamatergic synapses onto such NAc_D1R_ neurons to a KOR agonist. After establishing a baseline amplitude of AMPAR-mediated oEPSCs, they applied a KOR agonist to evaluate the effect on oEPSC amplitudes. They observed that the KOR agonist reduced amplitudes at BLA-NAc_D1R_ but not vHipp-NAc_D1R_ AMPAR-mediated synaptic responses. This effect was abolished by pre-application of a KOR antagonist. The presynaptic localization of the KOR was confirmed by demonstrating that specifically deleting the KOR from BLA neurons abolished the synapse-regulating effect of the KOR agonist at BLA-NAc_D1R_ synapses [80]. Likewise, another study used CRACM and pharmacological approaches to elucidate how opioid GPCRs regulate thalamocortical feedforward monosynaptic and disynaptic pathways [82]. They observed that monosynaptic thalamic inputs onto cortical pyramidal cells were directly under the inhibitory control of MORs. Instead, thalamic inputs impinging on cortical interneurons, providing thalamic disynaptic feedforward inhibition of cortical pyramidal cells, were under the inhibitory control of delta opioid receptors (DORs). Accordingly, DORs were observed at PV interneurons [82]. 

In addition, various studies have evaluated experience-dependent alterations in opioid GPCR control over synapses. In one example, opioid GPCR control over various inhibitory GABAergic inputs to the ventral tegmental area (VTA) of the rat was evaluated [83]. Optogenetic pathway specific activation of VTA inputs originating from either local interneurons, the rostromedial tegmental nucleus (RMTg) or the NAc were evaluated. Under basal conditions, MOR agonist DAMGO inhibited the GABAergic RMTg synapses onto VTA dopamine neurons much more strongly than it affected GABAergic NAc or local GABAergic VTA inputs onto dopamine neurons. Interestingly, when rats were administered morphine, there was less subsequent opioid-mediated inhibition at RMTg-VTA dopamine neuron synapses. This decreased sensitivity of the synapses to opioids after pre-exposure was hypothesized to contribute to drug tolerance [83]. Another study observed that bath application of MOR agonist DAMGO persistently suppressed excitatory transmission at corticostriatal, but not thalamocortical synapses where it had only transiently suppressive effects [84]. This MOR-driven long-term depression was modulated in an experience-dependent manner. Prior exposure to alcohol prevented the ability of DAMGO to both transiently and persistently suppress the corticostriatal synapses. Similarly, voluntary alcohol (but not sucrose) intake abolished the suppressing effects of DAMGO on these synapses [84].

#### 3.5.2. Summary and Alternative Strategies

Overall, the combination of CRACM with pharmacology is an important approach when it comes to unraveling how GPCRs regulate the function of specific synapses, and how that function is altered in the context of experience-dependent plasticity. When evaluating the locus of expression of GPCR-mediated effects, previously discussed metrics like PPR and 1/CV^2^ can be taken along. Interesting technological developments also permit the optical study of (opioid) GPCRs in brain slices in different manners. For instance, in recent years multiple light-activated chimeric GPCRs have been developed, including for MOR [81]. This opto-MOR has the light-sensitive portion of rat-derived rhodopsin, combined with the intracellular loops and C-terminus of rat-derived MOR. When opto-MORs are expressed in periaqueductal gray neurons, this results in light-driven barium-sensitive outward GIRK currents, comparable to those induced by endogenous opioids acting via the endogenous MOR [81]. Though not identical to a MOR, opto-MORs may give rise to exciting new avenues to explore GPCR control over synaptic function. Another interesting option with regard to the optical study of opioid control over neurotransmission, are photoactivatable neuropeptides. For instance, there are photoactivatable analogs of enkephalin (the endogenous agonist for MOR and DOR) and dynorphin (the endogenous KOR agonist), which can be optically uncaged to render them active. In a brain slice context, these compounds can be added to the ACSF and uncaged to induce local opioid receptor mediated transmission in neurons [85]. 

## 4. Dual Color Optogenetics for Synapse Interrogation

For a long time optogenetic approaches were largely limited to the study of one presynaptic input source onto postsynaptic neurons. The ability to stimulate multiple presynaptic inputs with distinct wavelengths in the same slice (Figure 3A), has evident advantages in terms of neural circuit dissection. For instance, it allows determining whether one neuron receives converging input from multiple sources [86,87,88]. It also allows determining synaptic temporal/spatial integration properties from various specific inputs [89]. To accomplish such dual color optogenetics technically, sufficient separation is required in the activation spectra of the excitatory opsin pair, which remains a challenge to this day, as we will discuss below. Chrimson is an example of an excitatory opsin with a red-shifted activation spectrum (peak response at 590 nm, with sensitivity over at least the 470–660 nm range) [90]. Another commonly used red-shifted opsin used in dual optogenetic studies is ReaChR (peak response at 590 nm, with sensitivity at least over the 410-650 nm range) [91]. These activation spectra differ considerably from those of ChR2 (peak response at 470 nm, with sensitivity over the <400–530 nm range). It also differs from Chronos, a ChR2 variant that has higher light sensitivity and faster kinetics than ChR2. Chronos has a peak response at 500 nm and sensitivity over at least the 470–530 nm range [90]. 

From the indicated activation ranges it is evident that the red-shifted activation spectra of Chrimson and of ReaChR still have partial overlap with the activation spectra of both ChR2 and Chronos, particularly when it comes to the use of blue light (Figure 3B). Therefore, it remains a challenge to ensure that when working with opsin pairs like ChR2-Chrimson, Chronos-Chrimson, or ChR2-ReaChR there is no meaningful cross-over: unintentionally co-activating the ‘wrong’ opsin. Specifically, finding red-shifted opsin-compatible wavelengths (e.g., >600 nm) that activate neither ChR2 nor Chronos is easier than finding ChR2/Chronos-compatible wavelengths that do not co-activate red-shifted opsins like Chrimson and ReaChR [90,91]. As the risk for meaningful cross-over diminishes when lower light intensities are used to stimulate opsins, Chronos (despite having a somewhat more red-shifted activation spectrum than ChR2) has an advantage over ChR2 in that it has higher blue light sensitivity, permitting larger current influx at low irradiance. For instance, in cultured neurons blue irradiances in the 0.05–1 mW/mm^2^ range were much more capable of driving large inward currents and spikes in Chronos-expressing cells than in ChR2-expressing neurons [90]. In practice both ChR2 and Chronos (and variants) are used in dual opsin studies as the counterpart to opsins like Chrimson or ReaChR, to assess the function of multiple synapses in the same brain slice [13,82,86,88,89,90]. 

### 4.1. Using Low (Blue) Irradiance to Obtain Specificity with Dual Color Optogenetics

#### 4.1.1. Using the Chrimson and Chronos Opsin Pair for Dual Color Optogenetics 

The original study discovering Chrimson and Chronos, also assessed their usability in brain slices for dual color optogenetics [90]. Mouse cortical neurons expressing either Chronos or Chrimson were patched in brain slices. LED-driven blue light stimulation and red-shifted light stimulation were assessed at different irradiances to examine their specificity in depolarizing the somata expressing the appropriate opsin. LED-driven blue light stimulation (470 nm peak) in a bandwidth irradiance of 0.2–0.5 mW/mm^2^ (i.e., low intensities) depolarized Chronos-expressing cortical neurons, but also limitedly depolarized Chrimson-expressing somata. The other way around, LED-driven red-light stimulation (625 nm peak) over a large irradiance bandwidth of 1–6 mW/mm^2^ depolarized Chrimson-expressing cells without affecting Chronos-expressing cells [90]. The authors then evaluated whether at the synaptic level sufficient specificity could be observed. They patched non-opsin-expressing cortical neurons near to Chronos- or Chrimson-expressing ones. In the patched cells postsynaptic currents could be evoked by Chronos activation (0.3 mW/mm^2^_,_ 470 nm, 5 ms light pulses by LED) and by Chrimson activation (4 mW/mm^2^, 625 nm, 5 ms pulses by LED). Importantly, in control conditions where just one of these opsins had been expressed, the aforementioned light conditions resulted in specific activation of the targeted opsin. In this study Chrimson-Chronos dual optogenetics could thus be used to examine synaptic function without cross-over, by using low blue light intensities (Figure 3B). While free of spectral cross-over, the synaptic current amplitudes were relatively small [90]. 

Another recent study used Chronos with ChrimsonR (a variant of Chrimson with a similar activation spectrum but with faster kinetics, allowing more fidelity at frequencies above 20 Hz than Chrimson itself [90]) to perform dual optogenetic interrogation of NAc circuits [13]. In this study various glutamatergic inputs (mPFC, vHipp, BLA, PVT) to Nac_D1R_ neurons were evaluated with dual optogenetics. The study used alternating blue light (0.2–3 mW, 470 nm, 5 ms light pulses by LED) and orange light (0.2–3 mW, 595 nm, 5 ms light pulses by LED) to stimulate synapses input-specifically. Control experiments were done in the presence of only one opsin, showing that orange light stimulation (595 nm, 1 mW) resulted in the absence of synaptic responses when only Chronos was present in mPFC, whereas some small EPSCs remained to blue light stimulation (470 nm, 1 mW) when only ChrimsonR was present in mPFC. The authors used low intensity stimulation of two independent pathways. Per animal Chronos was expressed either in the vHipp or in the BLA, and ChrimsonR in either the mPFC or in the PVT. The authors used dual color optogenetics to show that NAc_D1R_ cells often received converging inputs from mPFC and BLA (44%) or PVT and vHipp (55%) [13].

#### 4.1.2. Using the Chrimson and ChR2 Opsin Pair for Dual Color Optogenetics

There are also examples of studies using Chrimson and ChR2 (rather than Chronos) as an opsin pair to evaluate multiple synaptic pathways in brain slices [82,89]. In one such experiment convergent inputs onto the NAc were studied from the mPFC and BLA. To this end ChR2 was virally expressed in the mPFC and/or Chrimson was expressed in the BLA. When patching from NAc neurons in mice in which only ChR2 had been expressed in the mPFC, blue light stimulation (0.5–1 mW, 445 nm, laser) evoked sizeable glutamatergic synaptic currents, whereas red light stimulation (0.5–1.0 mW, 635 nm, laser) did not. Conversely, when patching from NAc neurons from mice in which only Chrimson had been expressed in the BLA, red light stimulation evoked sizeable EPSCs. Instead blue light at 0.5 mW evoked very small EPSCs of only a few picoamperes in size, whereas 1 mW stimulation yielded larger ‘off target’ responses. This scenario is in accordance with Chrimson still being sensitive to blue light at even low to moderate light intensities [90]. The authors therefore limited unintended blue light-driven Chrimson activation by using blue light intensities even below 0.5 mW (0.05–0.2 mW range) in follow-up experiments. Amongst these, they examined if NAc neurons performed synaptic integration on coinciding synaptic inputs from various input sources [89]. This illustrates the type of measure that is beyond the scope of sequential single color optogenetic experiments. 

### 4.2. Using Protracted Orange Light to Reduce Red-Shifted Opsin Sensitivity to Subsequent Blue Light

One strategy sometimes taken in dual optogenetics experiments, intending to mitigate the risk of the red-shifted-expressing neurons responding to the blue light stimulation, is to sequentially assess (synaptic) responses to a short flash of blue light after a longer pulse of orange/red light. This longer orange/red light pulse first activates the red-shifted opsin-expressing axons, but then the continuous stimulation inactivates the terminals, making them unable to partake in subsequent blue light driven responses (Figure 3C) [87]. This rationale was taken in a study demonstrating that ReaChR-expressing cortical neurons were excited both by blue light (~2 mW/mm^2^, 470 nm, LED) and by orange light (~2 mW/mm^2^, 590 nm, LED). Instead ChR2-expressing cortical neurons were excited only by the blue, but not the orange light [87]. The authors then demonstrated that first applying a long orange light pulse (50–250 ms) and then the blue light pulse (50 ms), resulted in blue light-driven (synaptic) responses that were exclusively ChR2-mediated. Specifically, the orange light pulse drives ReaChR-expressing terminal activation at the onset of the pulse, but at the time of its offset the ReaChR-expressing axons are temporarily inactivated and do not respond to the blue light, creating a temporal window where blue light responses are solely mediated by ChR2-expressing terminals [87]. 

This same strategy has been used for the ChR2-Chrimson opsin pair [86,88]. One study sought to determine whether patched GABA neurons in the lateral hypothalamus receive input from two separate ventral striatal inputs: the NAc and the VP [88]. NAc neurons were made to express Chrimson, whereas ventral pallidal neurons were made to express ChR2. Upon patching a hypothalamic neuron, a 250 ms pulse orange light pulse was delivered to first activate and subsequently inactivate Chrimson-expressing axons (605 nm, LED). Indeed, for the NAc somata, such a long pulse resulted in an initial spike at the onset, though not continuous spiking. If at the onset of this long light pulse there was a synaptic response, this was considered evidence for Chrimson-mediated synaptic activation (i.e., NAc-mediated). At the end of the 250 ms orange light pulse, an immediate stimulation to activate ChR2 followed (470 nm LED stimulation, 25 ms pulse). If this blue light pulse triggered a synaptic response, it was considered evidence for ChR2-mediated synaptic activation (i.e., VP-mediated). With this approach the authors showed that a considerable portion of GABAergic lateral hypothalamic neurons received converging VP and NAc GABA_A_R-mediated input [88]. 

Overall, the use of inactivating longer pulses of orange light to temporarily remove contributions of red-shifted opsin expressing axons to blue light, is an interesting strategy to achieve dual color optogenetics specificity. It is particularly useful when the research questions at hand can be addressed by stimulating inputs sequentially. For instance when addressing whether a neuron type receives converging synaptic input from different sources. 

### 4.3. Summary and Caveats of Dual Color Optogenetics

Overall, dual color optogenetics is feasible, though there are challenges in ensuring that the red-shifted opsin has negligible sensitivity to the blue light stimulation utilized. Typically, this issue is (partially) resolved by either using low blue irradiances (e.g., below 0.5 mW/mm^2^; Figure 3B) or by using long inactivating orange light pulses (to attempt to inactivate the red opsin expressing terminals) prior to the blue light stimulation (then irradiances can exceed 0.5 mW/mm^2^; Figure 3C). These different approaches both come with advantages and disadvantages in the type of experiments they permit. While the low blue irradiance strategy offers flexibility in terms of the experimental options it allows for, it comes at the cost of likely stimulating the blue opsin expressing axons with sub-optimal strength. This could make it easier to miss the existence of more sparse connections. Instead, the approach with protracted light pulses to inactive red-shifted terminals allows for higher stimulation intensities. This also confers experimental flexibility, as the recorded synaptic currents will generally be larger. However, with this approach one cannot determine processes of synaptic integration from divergent inputs, for which the input onsets would need to co-occur in close temporal proximity. Moreover, this approach requires substantial activation of the red opsin-expressing afferents in order to be able to evaluate the blue opsin-expressing afferents with specificity. This could yield a physiological crosstalk, for instance if the red-shifted opsin expressing afferents regulate the blue-shifted opsin expressing ones, which may be undesirable in certain experimental settings. Moreover, the strategy only works if the red-shifted opsin expressing neurons indeed become inactivated over the course of the long orange light pulse, which will be a cell-type specific process, [87] and will need to be experimentally verified. 

Furthermore, it should be kept in mind that there are no fully equivalent opsins in terms of relevant biophysical properties. For instance, Chronos may be advantageous over ChR2 as an opsin more easily stimulated with low blue light irradiances, but it also has considerably faster channel closing kinetics and higher fidelity at frequencies beyond 20 Hz compared to opsins like ChR2, Chrimson or ReaChR [90,91]. Moreover, there can be stochiometric differences in how many opsins will actually be expressed at the afferents. This may be both because of differences in stereotactic viral targeting, but also due to opsin-specific differences [7]. If dual color optogenetic strategies are used, it is therefore not only essential to find experimental conditions in which there is negligible spectral cross-over, but it is also advisable to consider inverting the opsin pair in control experiments to see if observations hold. 

## 5. Discussion

Here, we have discussed not only advantages but also general caveats and limitations of the individual synaptic metrics that can be taken with CRACM approaches. More generally, it is important to consider that the use of opsins to study neural circuits also comes with its own unique challenges. In part those pertain to the ways in which neurons are made to express the opsins. 

### 5.1. Considerations when Expressing Opsins

There are multiple ways in which opsins can be expressed in brain cells. One strategy involves the use of transgenic animals that themselves express opsins in particular cellular populations. This in turn can be achieved in various manners. One is to have transgenic animals in which promoters that are only active in specific cell types directly drive the expression of an opsin transgene. For instance, Vglut2-ChR2-YFP mice, in which opsin expression occurs in subtypes of glutamatergic cells (i.e., those with an active Vglut2 promoter) [92]. Alternatively, a binary system can be used in which in one transgenic line expression of the opsin is under the control of a strong ubiquitous promoter, but with expression dependent on a normally absent driver element (e.g., Cre recombinase). Such a mouse line can be crossed with a driver line, in which the expression of the driver element only occurs in specific cell types (e.g., a Vglut2-Cre mouse line). In that case the offspring of these mice have opsin expression in the specific cell type. These different approaches come with their own technical caveats, and we direct the interested reader to further discussion on this topic [92]. A different strategy altogether is to inject viral vectors (often stereotactically in the brain) to drive the expression of opsins in cells of interest. This can be either done in wildtype animals to drive direct opsin expression in cells of the targeted brain region. Instead, also here a binary approach is possible, where the injected viral vector used mediates opsin expression in a driver element-dependent way. The driver element can come from various sources, like other injected viral vectors or as the product of a transgenic animal (e.g., a Vglut2-Cre mouse). There is a multitude of viral vectors that are used for opsin expression approaches. A detailed discussion of their advantages and disadvantages goes beyond the purposes of the current review, but also here we direct the interested reader to discussions on the topic [5,93].

Of all the different strategies to express opsins in cells, the most common one in practice is the use of stereotactically injected AAVs. AAVs have considerable advantages such as generally favorable tropism when it comes to infecting brain cells, relatively low toxicity, and the possibility for the expression of the viral payload to be driver element-dependent (e.g., Cre-dependent). Most AAV serotypes infect the soma of an injected brain region and also achieve transgene expression in axons and nerve terminals. Notable is that AAV genomes do not integrate in the host cellular genomes, which can be a limitation in mitotic systems (e.g., the developing brain). Another clear disadvantage of AAVs, as compared to for instance lentivirus, is its limited maximal genome size. There are also considerable differences between AAV serotypes and variants which can differ substantially in transduction efficiency and specificity. Importantly, several AAV serotypes are associated with cytotoxicity, in particular at high doses [5,93].

Which AAV serotype is used can be a relevant factor when doing CRACM studies. For instance, one study assessed whether AAV-delivered ChR2-mediated optogenetic stimulation of particular central synapses revealed different properties, compared to studying these synapses with electrical stimulation [56]. Trains of 10 stimuli (electrically or optogenetically) were delivered at various frequencies. At cerebellar Purkinje cell synapses onto deep cerebellar nuclei, synaptic properties in terms of PPR over the course of the train (10 or 50 Hz) were highly similar across stimulation modalities. This was not the case however at hippocampal CA3-CA1 connections, where optogenetically probed synapses exhibited lower PPR values over the course of the train than with electrical stimulation. This was particularly the case for pulses after the 2nd pulse in the train. For PPRs calculated based on 2 pulses (Box 5), inter pulse intervals in the 50–500 ms range gave comparable PPR between optogenetic and electrical stimulation, whereas intervals shorter than 50 ms (e.g., 10 ms or 20 ms) gave much stronger depression with optogenetic stimulation. The authors assessed whether the potentially altered synaptic functionality was due to the opsin itself or due to the AAV-mediated viral delivery. They observed that in transgenic mice expressing ChR2 congenitally (without viral transfection) optogenetically probed CA3-CA1 synapse performance over 10 pulse trains of 10 or 50 Hz trains was identical in terms of PPR over the 10 pulses, compared to when the synapses were stimulated electrically. Since this finding pointed at the synapse-specific alterations potentially being due to AAV delivery, the authors examined different AAV serotypes. They observed that at CA3-CA1 synapses, AAV9 mediated delivery of ChR2 (as compared to AAV5 and AAV8) performed most like electrically stimulated synapses over 10 and 50 Hz trains. Even then, AAV9 performed better at hippocampal CA3-CA1 synapses when optical stimulation targeted the axons rather than directly the nerve terminals. The latter led to stronger synaptic depression, potentially due to light-driven ChR2-mediated influx of calcium directly into the terminal at levels beyond those that would be reached by an incoming action potential. The general mechanisms behind serotype-driven synapse-specific effects remain unresolved. One potential factor is the extent to which the injected serotype triggers reactive astrocytosis in the brain region, thus altering synaptic function [56]. 

### 5.2. The Bystander Effect 

Another interesting potential side effect with optogenetics in brain slices is the so-called ‘bystander effect’. In this case, a non-opsin expressing neuron surrounded by (many) opsin-expressing ones, could be influenced by changes in the extracellular ionic milieu when optogenetic stimulation drives ion flow across membranes of the many surrounding opsin-expressing neurons [94]. In one study such effects were shown by using AAV-mediated unilateral delivery of ChR2 in the hippocampus, which leads to contralateral projections. The authors then patched a hippocampal neuron on the contralateral side in the presence of synaptic blockers. With very long light stimulation (i.e., a 15 s blue light pulse at 5 mW/mm^2^), the patched neuron expressed an inward current (~150 pA), a decreased membrane resistance, and a 6 mV depolarization. Additionally, pulsed trains of 20 Hz or 10 Hz over 15 s could produce tens of picoamperes of slow-kinetic inward bystander currents [94]. This indicates that in specific circumstances very strong optogenetic stimulation may also influence nearby cells that are not part of the network under investigation. 

### 5.3. Troubleshooting by Limiting Viral Loads and Light Intensities 

To remedy many of the potential issues with CRACM approaches, it is important to appropriately choose the properties of the opsin(s) used, to titrate down both viral titer loads and volumes used [5], and to avoid unnecessarily long or intense light stimulation pulses. With regard to modalities of light delivery there are also flexible possibilities. Much of the work discussed here used wide-field illumination of tissue, via LED-driven stimulation. Wide-field stimulation has strong points, such as relative consistency in intensity and location of optical stimulation [66]. However, more spatially restricted forms of stimulation can confer great specificity. For instance, sub-cellular CRACM techniques (sCRACM) entail the application of laser-beam stimulation points over a virtual grid over a labeled neuron, to identify where the presynaptic terminals reside with respect to the morphology of the patched cell [32]. Other combinations with advanced optics allow for detailed exploration of circuit properties. For instance, application of two photon approaches allow for holographic stimulation of geometrical shapes in brain slices. In this case, individual neurons can be separately targeted while leaving their neighbors unstimulated, when they express somatically restricted opsin forms [95]. 

### 5.4. Concluding Remarks

Here, we reviewed the state of the field of ex vivo optical interrogation of synaptic connectivity and strength. We aimed to offer a critical view to current methodologies and to provide practical insights when using these approaches. As we have discussed, the combination of optogenetics and patch-clamp electrophysiology in brain slices has proven to be an indispensable tool in neuroscience. It permits the unraveling of the functional architecture of neural circuits, and the determination of the effects of salient environmental experiences such as reward and stress on specific synaptic functioning. Across several decades of use, despite its limitations, the future potential for optogenetics with patch clamp electrophysiology in neural circuit dissection remains bright.

## Figures and Tables

**Figure 1 ijms-23-11612-f001:**
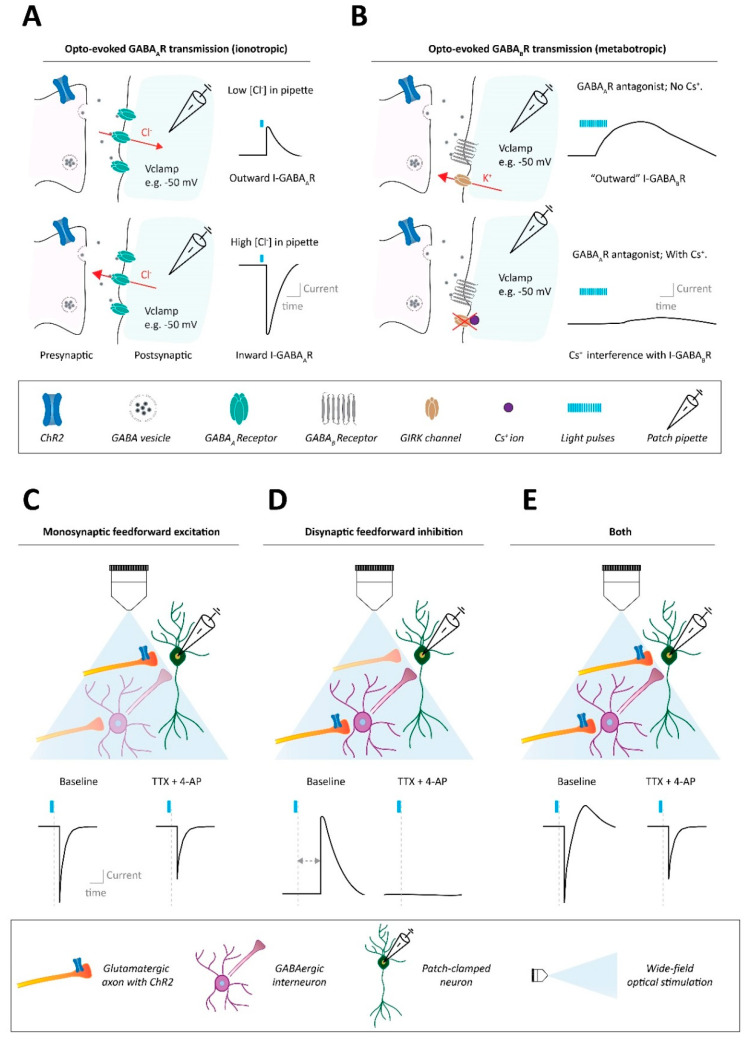
Methods to assess synaptic connectivity in neural circuits. (**A**) Ionotropic synaptic transmission assessed with optogenetic approaches, using single pulse stimulation. Example of GABA_A_R synaptic transmission. Top: Using a patch pipette solution with a low-chloride ion concentration results, at slightly depolarized voltage clamping holding potentials such as −50 mV, in outward GABA_A_R transmission (i.e., Cl^−^ influx through GABA_A_Rs) after optogenetic stimulation of GABAergic axons. Bottom: Instead, using a patch pipette solution with a high-chloride ion concentration results in inward GABA_A_R transmission (i.e., Cl^−^ efflux through GABA_A_Rs) after optogenetic stimulation of GABAergic axon terminals. (**B**) Metabotropic synaptic transmission assessed with optogenetic approaches, stimulating with trains of pulses. Example of GABA_B_R synaptic transmission. Top: When using a patch pipette potassium-based solution (no cesium), trains of optogenetic stimulation, at slightly depolarized voltage clamping holding potentials such as −50 mV, can induce slow kinetic outward GABA_B_R transmission (i.e., K^−^ efflux through GABA_B_R-regulated potassium channels like GIRKs. Bottom: When using a patch pipette solution with a high cesium internal content, the cesium ion blockade of potassium channels like GIRKs interferes with the ability to detect metabotropic GABA_B_R signaling after optogenetic GABA terminal stimulation. (**C**) Whether an optogenetically evoked synaptic input onto a patched neuron is monosynaptic can be determined with the pharmacological combination of tetrodotoxin (TTX) and 4-aminopyridine (4-AP). In the presence of TTX, monosynaptic feedforward input onto the patched neuron should be salvageable by 4-AP administration. (**D**) Instead, if optogenetic stimulation indirectly leads to an observed synaptic current into the patched cell (e.g., an outward current due to disynaptic inhibition), such a polysynaptic current will not be observed after TTX + 4-AP application. (**E**) Optogenetic stimulation may lead to combinations of mono- and polysynaptic activity onto the patched neuron, which can also be separated via the combination of TTX + 4-AP.

**Figure 2 ijms-23-11612-f002:**
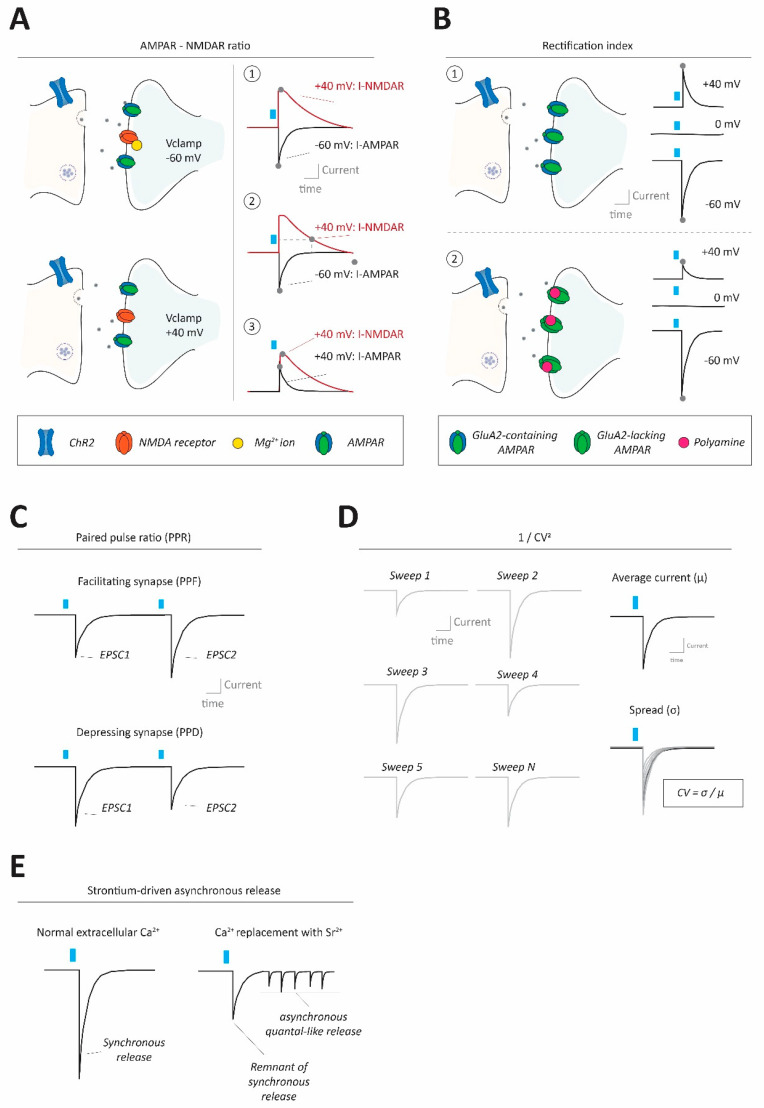
Methods to assess synaptic strength and properties in neural circuits. (**A**) Using optogenetic stimulation at excitatory synapses AMPA-NMDA ratios can be calculated to gauge synaptic strength, by normalizing AMPAR-mediated to NMDAR-mediated currents (I-AMPAR; I-NMDAR). This can be done in different ways, requiring conditions to separate the two currents from each other. A_1_: Clamping the cell at −60 mV allows for determination of I-AMPAR at peak values (without I-NMDAR contributions due to its block by Mg^2+^ ions under these conditions, top left), and then clamping the cell at +40 mV allows for determination of I-NMDAR peak values (alleviating Mg^2+^ block, bottom left). In this case, the I-NMDAR value will require pharmacological separation from I-AMPAR. A_2_: Instead of using pharmacological separation between I-AMPAR and I-NMDAR at +40 mV, it is also possible to evaluate I-NMDAR at a delayed timepoint after stimulation (e.g., 100 ms), where I-AMPAR has decayed, but I-NMDAR has not due to its slower kinetics. A_3_: I-AMPAR and I-NMDAR can also both be taken, at peak values, at +40 mV requiring pharmacological isolation of one of the two components to separate them. (**B**) Optogenetic interrogation of input-specific AMPAR subunit composition (changes) can be done by assessing current/voltage (I/V) relationships of the receptor. Top: Typical AMPARs, containing RNA-edited GluA2 subunits, conduct equally well inwardly (i.e., net cation influx into the cell) as they do outwardly (i.e., net cation efflux out of the cell). Thus, with a reversal potential at 0 mV, those AMPARs would conduct about 1.5× (60/40) more inward current at −60 mV compared to +40 mV. Bottom: GluA2-lacking AMPARs (which can conduct calcium) instead exhibit inward rectification (i.e., larger inward currents than outward currents) mainly due to polyamine block of the receptor at depolarized membrane states under which outward currents would occur. (**C**) Optogenetics can be used to evaluate the synaptic paired pulse ratio (PPR), often indicative of presynaptic release properties. PPR is calculated by giving two (or more) pulses in quick succession, generating two separate postsynaptic currents, and dividing the amplitude of the second by the first response. PPRs can reveal whether synapses are facilitating (paired pulse facilitation; PPF; top) or depressing (paired pulse depression; PPD; bottom). (**D**) The coefficient of variance (CV), often reformulated as 1/CV^2+^ for practical reasons, can be informative about presynaptic properties such as release probability and number of release sites. CV itself is calculated by dividing the standard deviation of postsynaptic amplitudes by the mean of the amplitudes. (**E**) Under conditions where ACSF calcium ion concentration is replaced by an equimolar (or higher) concentration of strontium ions, synaptic transmission becomes more asynchronous and ‘quantal-like’. Optogenetic synapse stimulation under strontium conditions allows approximation of the quantal size at specific synapses.

**Figure 3 ijms-23-11612-f003:**
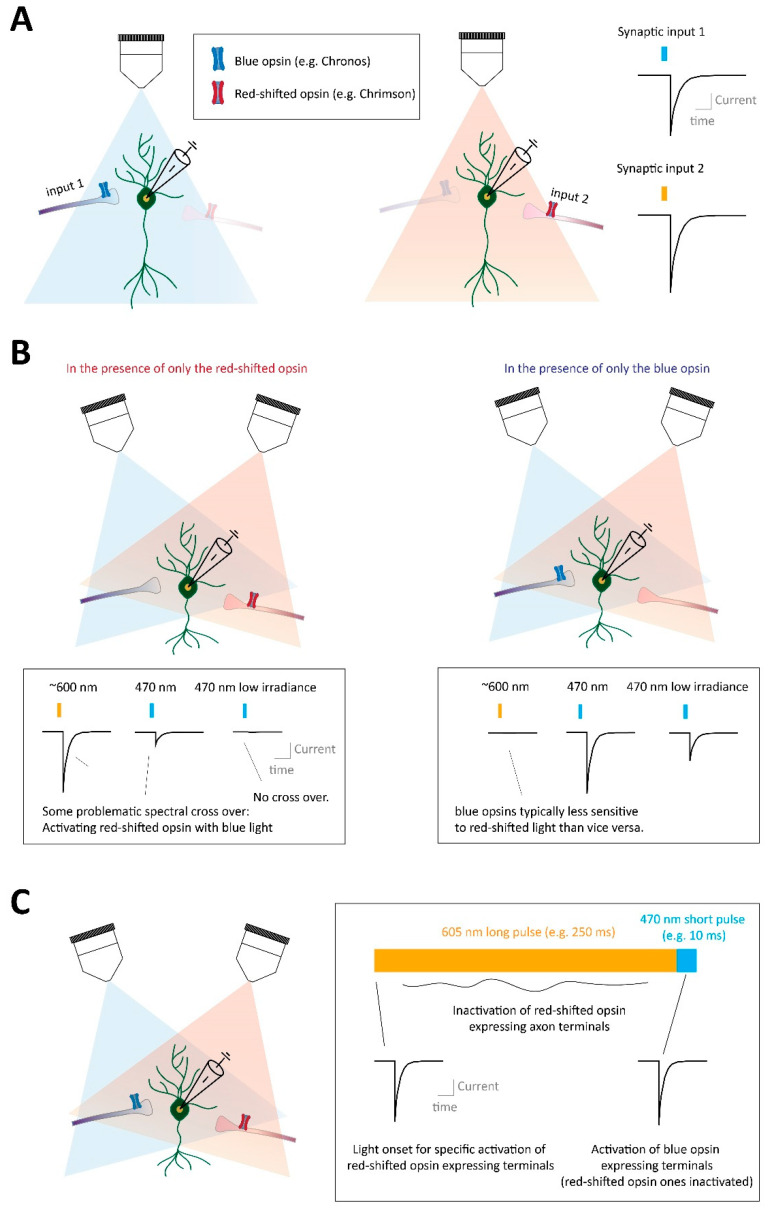
Dual color optogenetics to assess properties of multiple synapses in a neural circuit. (**A**) When combining expression of a blue light sensitive opsin (e.g., ChR2 or Chronos) with a more red-shifted opsin (e.g., Chrimson) in distinct presynaptic sources, it becomes possible to stimulate separate synaptic inputs in a slice, by stimulating the tissue with blue light wavelengths vs. red-shifted ones. (**B**) Control experiments are required to assess the extent of spectral cross over. Left: A situation where only the red-shifted opsin (Chrimson) has been expressed, and the tissue is stimulated with either blue or orange wavelengths. Orange wavelengths indeed activate the axon terminals causing synaptic responses. However, medium to strong blue light irradiance will also (unintentionally) stimulate the red-shifted Chrimson. Instead if low irradiance blue light is used, this does not sufficiently activate Chrimson and no ‘off-target’ synaptic transmission occurs. Right: The opposite scenario where only the ‘blue light’ opsin is expressed (Chronos). Typically orange light stimulation will not activate this opsin and no ‘off-target’ synaptic transmission occurs. Instead, the opsin has sensitivity to blue light, even with low irradiance. Thus, with low blue light stimulation intensities, spectral crosstalk can be minimized. (**C**) An alternative approach to separate contributions of (left) blue opsin-expressing vs. Chrimson-expressing nerve terminals, without minimizing stimulation intensities. Right: A prolonged pulse of orange light (~605 nm) will typically activate (pulse onset) but then subsequently inactivate (during pulse) Chrimson-expressing nerve terminals. Thus, if at the offset of the orange light pulse (with Chrimson-terminal inactivation) a blue light pulse is given, it activates ‘blue opsin’ expressing terminals without concurrent contributions of the (blue-light sensitive) Chrimson expressing ones.

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
