# Peer review of "Studying Synaptic Connectivity and Strength with Optogenetics and Patch-Clamp Electrophysiology"

_ijms, 2022, doi:10.3390/ijms231911612_

Round 1

Reviewer 1 Report

This is a nice and useful beginner’s introduction into the topic. This review will help students to understand the rationale for the design of the type of experiments covered.  A minor concern is that CRACM is described without discussing some caveats and concerns (synaptic properties are likely not physiological if presynaptic calcium transients are induced directly by optogenetic stimulation). Another minor suggestion would be to state upfront the main issue of dual-colour optogenetic stimulation approaches, namely the non-negligible overlap of activation spectra of “blue” and “red-shifted” opsins.

I like to note (not required for acceptance) that a more advanced course would require a higher precision in the descriptions of the biophysical underpinnings of the phenomena discussed and departure from the - implicit - concept that what has been previously done and published is the best possible forward going experimental design option.

Author Response

Thank you for your comments. Please see the attachment for our answers to the raised points.

Reviewer 2 Report

In this review, the authors gave an overview of how ChR2-assisted circuit mapping (CRACM)-like methods have been applied to understand various parameters of neural circuit function. First, they explain how CRACM-like methods help determine the synaptic circuits involving ionotropic or metabotropic receptors. They then argued the different synaptic functionality measurements. Finally, they reviewed the use of dual color optogenetics to study different neural circuits.  Most importantly, they also discussed potential caveats and limitations in these approaches. 

Overall, it looks that this review is extensively and carefully written. Those practical and precise information will be helpful to many readers, although I felt that some points could be more succinct. Thus, I do not suggest deleting anything from the current version.  I did not check the many values shown by the authors one by one.  I would request the copy-editors of this journal to carefully take care of adjusting the format of this manuscript andfor the authors to closely look at the final proofs.   

The authors seem to have much experience with recordings from brain slices.  However, I woud suggest adding the following aspects to further broaden the readership of this excellent review.  

1) In most cases, the authors intend to use AAV vectors in their approaches.  The AAV tools are commercially procured, produced in core facilities, or readily available from other researchers. However, it is also important to consider what kind of AAVs (or lentivirus vectors) are used and how the optogenetic molecules are overexpressed (eg, serotype, promoter).  In mice, it is also practical to use the loxP/Cre system using AAV or crossing various genetically modified transgenic lines.  It is also critical to appropriately select these tools to set up CRACM-like approaches. The wrong choice can also lead to caveats in some occasions (eg. low expression level, cross over, toxicity due to too much expression). Although the authors briefly discussed the related problem (line 1023 onwards), the authors should mention these issues by citing more papers. 

2) The authors probably intend to use mice or rats in this review. However, CRACM-like approaches can be used in other animals (eg, Drosophila, zebrafish, and non-human primates).  The authors might want to mention that the points discussed in this review are also valuable to various cases in other species. 

Author Response

(The authors gave the same response as above.)
